# Hyperacetylated histone H4 is a source of carbon contributing to lipid synthesis

Evelina Charidemou[1], Roberta Noberini[2,3], Chiara Ghirardi[2,3], Polymnia Georgiou [ID][1], Panayiota Marcou[1], Andria Theophanous[1], Katerina Strati [ID][1], Hector Keun[4], Volker Behrends [ID][5], Tiziana Bonaldi [ID][2,3] & Antonis Kirmizis [ID][1][✉]

## Abstract

**Histone modifications commonly integrate environmental cues with cellular metabolic outputs by affecting gene expression. However, chromatin modifications such as acetylation do not always correlate with transcription, pointing towards an alternative role of histone modifications in cellular metabolism. Using an approach that integrates mass spectrometry-based histone modification mapping and metabolomics with stable isotope tracers, we demonstrate that elevated lipids in acetyltransferase-depleted hepatocytes result from carbon atoms derived from deacetylation of hyperacetylated histone H4 flowing towards fatty acids. Consistently, enhanced lipid synthesis in acetyltransferase-depleted hepatocytes is dependent on histone deacetylases and acetyl-CoA synthetase ACSS2, but not on the substrate specificity of the acetyltransferases. Furthermore, we show that during diet-induced lipid synthesis the levels of hyperacetylated histone H4 decrease in hepatocytes and in mouse liver. In addition, overexpression of acetyltransferases can reverse diet-induced lipogenesis by blocking lipid droplet accumulation and maintaining the levels of hyperacetylated histone H4. Overall, these findings highlight hyperacetylated histones as a metabolite reservoir that can directly contribute carbon to lipid synthesis, constituting a novel function of chromatin in cellular metabolism.**

**Keywords** Epigenetics; Histone Reservoirs; Lipid Metabolism; Acetylation
**Subject Categories** Chromatin, Transcription & Genomics; Metabolism; Post-translational Modifications & Proteolysis

## Introduction

In recent years, there has been great interest in understanding the multi-dimensional link between metabolism and epigenetics. A well-characterised aspect of this nexus relies on how metabolic inputs, including nutrient sensing from the diet, microbiome and immune responses, can alter the epigenetic landscape to regulate gene expression, linking the environment to the control of cellular homoeostasis and cell fate (Dai et al, 2020; Reid et al, 2017). The ability of metabolism to modulate the epigenome depends on the intrinsic property of chromatin-modifying enzymes to use metabolites as substrates to modify histones (Sabari et al, 2017; Haws et al, 2020). For instance, histone acetyltransferases (HATs) use acetyl coenzyme A (acetyl-CoA) to acetylate histones. Consequently, epigenetic modifications are sensitive to metabolic fluctuations that can arise from environmental or intracellular changes in nutrient availability (Tang et al, 2017; Fellows et al, 2018; Reina-Campos et al, 2019).

Many of these histone marks often do not correlate with transcriptional activity, indicating that they may contribute to chromatin-regulated cellular processes through alternative paths (Ye and Tu, 2018; Perez and Sarkies, 2023). This directed the scientific community towards a less studied aspect of the relationship between metabolism and epigenetics, where histone modifications could contribute directly to the regulation of metabolite pools (Ye and Tu, 2018; Wong et al, 2017; Katada et al, 2012; Lozoya et al, 2018). Specifically, histones can potentially store substantial amounts of acetyl groups, due to their high abundance in cells. In fact, the human genome can potentially harbour 4 billion acetyl-CoA molecules through histone acetylation (Ye and Tu, 2018), which is a modification with a high turnover rate (Jackson et al, 1975). Therefore, due to the relative activity of histone acetylation and deacetylation, it was hypothesised that histones could serve as a reservoir for acetyl-CoA, contributing to changes in acetyl-CoA pools (Nirello et al, 2022).

To assess the direct contribution of histone modifications to metabolite pools, it is important to characterise the competing flow of chemical groups from histone marks to metabolites in different physiological contexts (McDonnell et al, 2016; Mews et al, 2019; Nirello et al, 2022). Recent evidence supports this notion by demonstrating that acetyl-CoA is released from histones in the form of acetate and recycled within chromatin for histone acetylation at other genomic loci and other histone residues under certain metabolic shifts (Hsieh et al, 2022; Mendoza et al, 2022).

[1]Department of Biological Sciences, University of Cyprus, 2109 Nicosia, Cyprus. [2]Department of Experimental Oncology, IEO, European Institute of Oncology IRCCS, 20139 Milan, Italy. [3]Department of Oncology and Haematology-Oncology, University of Milano, Via Festa del Perdono 7, 20122 Milano, Italy. [4]Cancer Metabolism & Systems Toxicology Group, Division of Cancer, Department of Surgery and Cancer, Imperial College London, London, UK. [5]School of Life and Health Sciences, Whitelands College, University of Roehampton, London, UK. ✉E-mail: kirmizis@ucy.ac.cy

Acetyl-CoA lies at the intersection of the epigenetics-metabolism nexus, where histone acetylation competes for the same nucleo-cytosolic pools of acetyl-CoA with lipid synthesis (Galdieri and Vancura, 2012). There are two primary pathways that contribute to lipogenic acetyl-CoA pools; one is mediated by ATP citrate lyase enzyme (ACLY), which converts mitochondrial-derived citrate to acetyl-CoA (Zhao et al, 2016; Wellen et al, 2009), and the other through acetyl-CoA synthetase short-chain family member 2 (ACSS2), which typically synthesises acetyl-CoA from protein or histone deacetylation (HDAC)-derived acetate (Gao et al, 2016). Even though clinical trials are pursuing the inhibition of ACLY to stall lipid accumulation in metabolic disorders, evidence supports that during fructose-induced fat accumulation ACSS2-processed acetate supplies substantial lipogenic acetyl-CoA independently of ACLY (Zhao et al, 2020). In agreement, HDAC inhibition has been shown to alleviate fat accumulation in non-alcoholic steatohepatitis, proposing the involvement of deacetylation in lipid infiltration-related disorders (Huang et al, 2022).

The interplay between histone acetylation and lipid synthesis is exemplified by the fact that lipids become a major source of acetyl-CoA to alter histone acetylation (McDonnell et al, 2016). However, it remains unclear whether the reverse process occurs, whereby carbons from histone acetylation sites contribute towards lipid synthesis. Interestingly, we have previously shown that depletion of histone N-alpha acetyltransferase 40 (NAA40) leads to a substantial increase in acetyl-CoA levels and to a significant induction of lipid synthesis in hepatocytes. Notably, this metabolic rewiring upon NAA40 depletion preceded any transcriptional deregulation, and it was dependent on ACSS2 (Charidemou et al, 2022). In light of this, we hypothesised that upon reduction of acetyltransferase expression, histone deacetylation may have a direct contribution towards lipid synthesis.

Hence, in the present study, we aimed to investigate whether the depletion of various acetyltransferases, including both lysine acetyltransferases (KATs) and N-terminal acetyltransferases (NATs), could induce lipid synthesis in hepatocytes and to define the source of carbon for lipid accumulation upon their depletion. Using murine hepatocytes, we initially show that depletion of different acetyltransferases enhances lipid synthesis in a manner that is not dependent on their substrate specificity. Next, by employing MS-based histone epigenetic mapping and metabolomics with stable isotope tracer studies, we demonstrate that the elevated lipids in acetyltransferase-depleted cells resulted from carbon atoms flowing from the deacetylation of hyperacetylated histone H4 to fatty acids. This enhanced lipogenesis is driven by the common activity of acetyltransferases, including both histone and protein acetyltransferases, to consume acetyl-CoA and modulate hyperacetylated histone reservoirs by the action of ACSS2. Consistent with this, we show that during dietary fructose-induced lipid synthesis, a process that is known to be driven through acetate and ACSS2 (Zhao et al, 2020), there is a reduction of hyperacetylated histone H4 in cultured hepatocytes and mouse livers. Moreover, overexpression of acetyltransferases prevents fructose-induced lipid accumulation and maintains the levels of hyperacetylated histone H4. Overall, our findings introduce a novel link between epigenetics and cellular metabolism. Specifically, we show for the first time that histones, known primarily for their involvement in chromatin regulation, possess the ability to contribute as a significant source of metabolites, fuelling cellular processes such as lipid synthesis.

# Results

## Acetyltransferase depletion induces lipid accumulation in hepatocytes

Based on our previous work demonstrating that depletion of NAA40 induced lipid synthesis by affecting acetyl-CoA levels (Charidemou et al, 2022), we examined whether this applies to other protein acetyltransferases. We examined four acetyltransferases with varying degrees of estimated acetyl-CoA consumption and histone or protein target specificity (see Table 1 for calculations) (Wang et al, 2018; Shurubor et al, 2017; Zheng et al, 2013; Cieniewicz et al, 2014; Gupta et al, 2008; Milo, 2013; Hole et al, 2011; Scott et al, 2017; van Damme et al, 2011; Basile et al, 2019). A pool of four different *Myst-1*, *Gcn5*, *Naa40* or *Naa10* siRNAs efficiently depleted the mRNA levels of each acetyltransferase separately in AML12 hepatocytes, compared to cells treated with scrambled siRNAs (Fig. EV1A). Following a robust knockdown of each acetyltransferase in hepatocytes, we next examined their metabolome, lipidome and fatty acid profile, using HPLC-MS and GC-MS. Depleting acetyltransferases in hepatocytes did not lead to significant changes in key metabolic pathways, including glycolysis, the TCA cycle, nucleotide biosynthesis, the urea cycle, the methionine cycle and amino acid metabolism (Fig. EV1B). In contrast, the lipidome and fatty acid profile of acetyltransferase-depleted hepatocytes was significantly altered (Fig. 1A–C). Resulting lipid and lipid-bound fatty acid data were processed by constructing principal component analysis (PCA) models to identify the distribution of lipids and fatty acids among the MYST1-KD, GCN5-KD, NAA40-KD, NAA10-KD and scramble

**Table 1. Estimation of acetyl-CoA consumption by acetyltransferases examined in this study.**

| | |
|---|---|
| Genome size (bp) | $6.4 \times 10^9$ |
| Number of nucleosomes | $3 \times 10^7$ |
| Number of proteins (in hepatocytes) | $\sim 2.6 \times 10^7$ |
| Volume of cell (L) | $2 \times 10^{-12}$ |
| Intracellular [acetyl-CoA] (µmol/L) | 20 |
| **Number of acetyl-CoA per protein/nucleosome** | |
| NAA10 | $1–1.3 \times 10^{7}$[a] |
| NAA40 | $1.2 \times 10^{8}$[b,c] |
| GCN5 | $4.08 \times 10^{7}$[b,c] |
| MYST1 | $2.16 \times 10^{7}$[b,c] |
| **Consumable acetyl-CoA (µmol/L)** | |
| N-terminal acetylation by NAA10 | 9–11 |
| N-terminal acetylation by NAA40 | 100 |
| Lysine acetylation by GCN5 | 34 |
| Lysine acetylation by MYST1 | 18 |

[a]In total, 40–50% of cytosolic proteins in eukaryotic cells may undergo N-terminal acetylation by NAA10.
[b]Two acetyl-CoA molecules, one for each of the two H2A, H3 and H4 in a histone octamer.
[c]Acetylation abundance for different sites: NαAc is 98%; H3K14, 28%; H3K23, 25%; H3K9, 0.9%; H3K18, 4.1%; H3K27, 2.7% and H3K36; 6.9%; H4K16; 35.7%.

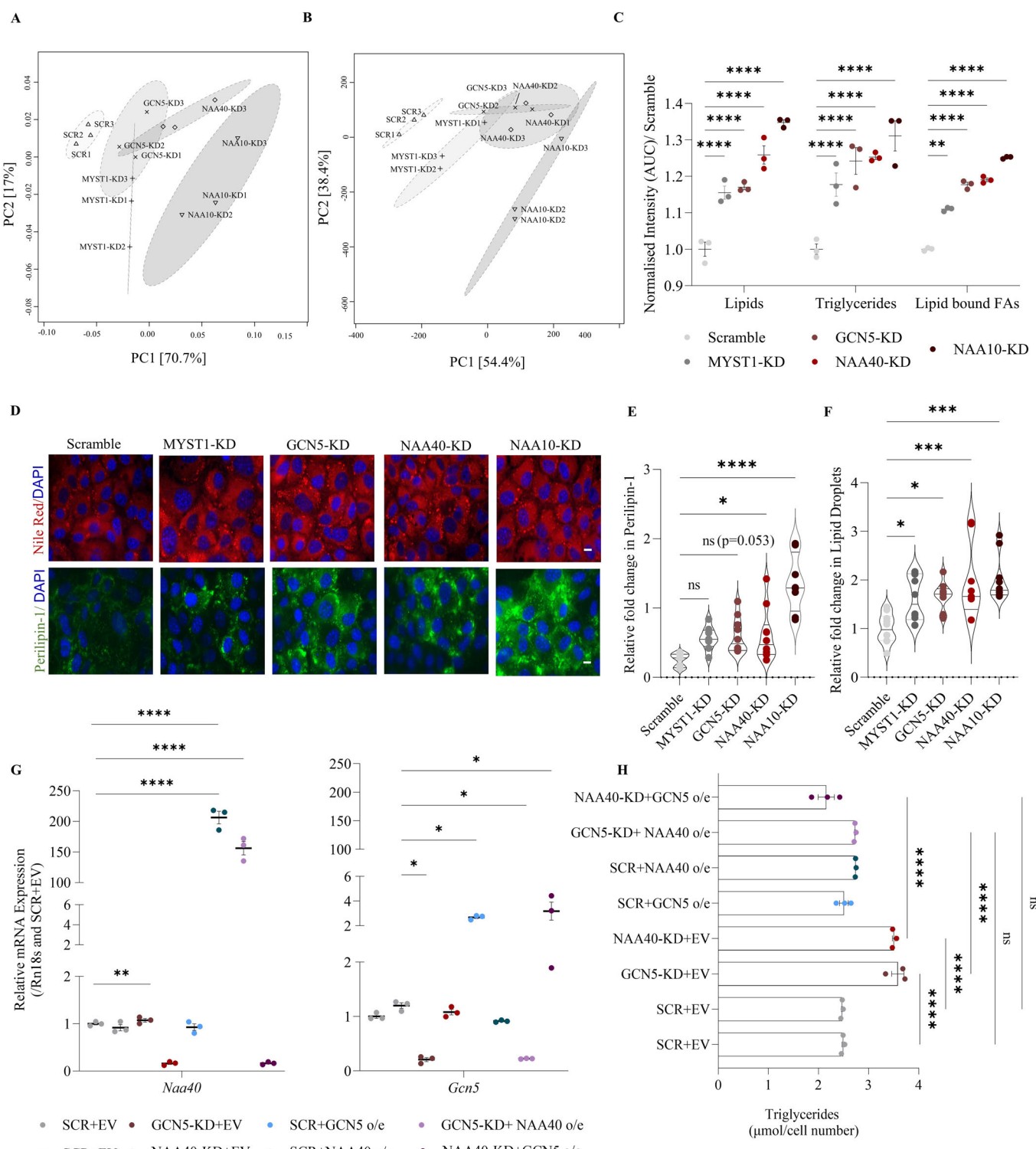

cells. PCA revealed that acetyltransferase-KD cells had a distinct lipid (Fig. 1A) and lipid-bound fatty acid profile (Fig. 1B) compared to scramble cells. Consistently, total lipids, triglycerides, and lipid-bound fatty acids were significantly increased in all four acetyltransferase-depleted cell models compared to scramble cells (Fig. 1C). In agreement with the increase in lipid accumulation, the levels of free fatty acids and cholesterol remained unchanged in

acetyltransferase-KD cells (Fig. EV1C), suggesting that there is no alteration in fatty acid breakdown, but rather enhanced lipid synthesis.

Lipids are commonly deposited as lipid droplets in cells and are typically coated by adherent proteins, such as perilipin-1, to regulate lipid homoeostasis (Okumura, 2011). In fact, increased hepatic lipid droplet content has been associated with steatosis and

**Figure 1. Acetyltransferase depletion increases lipids and fatty acids in AML12 hepatocytes.**

(A) PCA scores plot (PC1, 71.7% against PC2, 17%) showing separation in intact lipid species acids between the scramble, MYST1-KD, GCN5-KD, NAA40-KD, and NAA10-KD sample groups. PC1 is positively correlated with intact lipid content. $n = 3$ biological replicates/group. (B) PCA scores plot (PC1, 54.4% against PC2, 38.4%) showing separation in lipid-bound fatty acids between the scramble, MYST1-KD, GCN5-KD, NAA40-KD and NAA10-KD sample groups. PC1 is positively correlated with lipid-bound fatty acid content. $n = 3$ biological replicates/group. (C) Sum of total lipids, triglycerides, and lipid-bound fatty acids measured by MS in scramble, MYST1-KD, GCN5-KD, NAA40-KD and NAA10-KD cells after 48 h of siRNA treatment. $n = 3$ biological replicates/group. Statistical analysis was performed using two-way ANOVA with post hoc Tukey's multiple-comparisons test; **$P \le 0.01$, ****$P \le 0.0001$. (D) Representative image of lipid droplets by Nile red (red), Perilipin-1 (green), and nuclei by DAPI (blue) in sample groups 48 h after siRNA treatment; Scale bar = 25 µm. $n = 8$–10 biological replicates/group. (E) Quantification of relative lipid droplets in scramble, MYST1-KD, GCN5-KD, NAA40-KD, and NAA10-KD cells. $n = 8$–10 biological replicates/group. Statistical analysis was performed using one-way ANOVA with post hoc Dunnett's multiple-comparisons test; *$P \le 0.05$, ****$P \le 0.0001$, ns non-significant. (F) Quantification of Perilipin-1 using ImageJ in scramble, MYST1-KD, GCN5-KD, NAA40-KD, and NAA10-KD cells. $n = 8$–10 biological replicates/group. Statistical analysis was performed using one-way ANOVA with post hoc Dunnett's multiple-comparisons test; *$P \le 0.05$, ***$P \le 0.001$. (G) RT-qPCR analysis of expression of *NAA40* (left) and *Gcn5* (right) mRNA levels, after 48 h in represented treatment groups. $n = 8$–10 biological replicates/group. Statistical analysis was performed using one-way ANOVA with post hoc Dunnett's multiple-comparisons test; *$P \le 0.05$, ****$P \le 0.0001$. (H) Triglyceride levels after 48 h in represented treatment groups. $n = 3$ biological replicates/group. Statistical analysis was performed using one-way ANOVA with post hoc Dunnett's multiple-comparisons test; ****$P \le 0.0001$, ns non-significant. Data information: (C, G, H) are presented as mean ± SEM; (E, F) are presented as violin plots. Source data are available online for this figure.

insulin resistance (Greenberg et al, 2011; Zhang et al, 2013), whereby lipid droplets physically interact with stress kinases, inhibiting insulin signalling activation and blocking glucose uptake (Gassaway et al, 2020). Accordingly, all four acetyltransferase-depleted cell models had significantly increased content of lipid droplets (Fig. 1D,E), as well as increased levels of perilipin-1 (Fig. 1D,F) compared to scramble control cells. In addition, acetyltransferase-depleted cells had decreased levels of pAKT Ser473 (Fig. EV1D) and decreased glucose uptake (Fig. EV1E), which are indicative of reduced insulin signalling, validating the observed accumulation of lipid droplets.

To address whether the increase in lipids relates to the specific activity of each acetyltransferase or is an effect of overall reduced acetylation, we overexpressed GCN5 in NAA40-KD cells, or conversely, overexpressed NAA40 in GCN5-KD cells (Fig. 1G). Interestingly, overexpressing GCN5 in NAA40-KD cells or over-expressing NAA40 in GCN5-KD cells prevented the previously observed increase in triglyceride content in acetyltransferase-KD cells (Fig. 1H). This result suggests that the acetyltransferase-depletion effect depends on the overall reduced acetyltransferase activity and not on their substrate specificity and specific transcriptional outcomes. In fact, acetyltransferase depletion did not affect the expression of genes involved in lipid synthesis (Fig. EV1F). In support of this, the transcriptome of NAA40-KD cells at 12 h did not show significant enrichment of any biological process implicated in lipogenesis, when compared to control scramble cells (Fig. EV1G), while we have previously shown that lipid droplet formation is significantly induced at this time point following NAA40 depletion (Charidemou et al, 2022). Even though there were significant gene expression changes at 48 h post NAA40-knockdown, again these did not show enrichment in any metabolic processes implicating lipid synthesis (Fig. EV1H,I). These findings suggest that the increase in lipid content in acetyltransferase-depleted cells is not linked to their substrate specificity and subsequent transcriptional regulation but is likely due to their general activity of consuming acetyl-CoA.

## Acetyltransferase-dependent lipid accumulation is not associated with ACLY-derived acetyl-CoA

We next sought to determine the source of acetyl-CoA for enhanced lipid synthesis upon acetyltransferase deficiency. One possible source is exogenous, from glucose, through the activity of ACLY (Fig. 2A). To address this, we supplemented AML12 hepatocytes with $^{13}C_6$-Glucose upon siRNA-induced depletion of each acetyltransferase (Fig. 2A). We measured the level of $^{13}C$ incorporation in acetylated histone peptides by bottom-up MS analysis. The relative levels of $^{13}C$ were greater than $^{12}C$ in both histones and lipids (Fig. EV2A,B). The relative label incorporation ($^{13}C$) in mono- and di-acetylated histone peptides did not change among the different siRNA treatments, relative to the scramble control (Fig. 2B). The only observed change was an increase in the tri-acetylated form of histone H4 peptide H4K5K8K12K16 carrying a singly labelled acetyl-CoA $[(^{13}C_2)_1]$, which was increased in acetyltransferase-depleted cells compared to control (Fig. 2B,C). Of note, the tetra-acetylated form of the same peptide (4ac) could not be detected in this experiment. In addition, the labelled fraction of lipids was generally not increased in MYST1-KD, GCN5-KD, NAA40-KD and NAA10-KD cells compared to control, with instead a slight decrease for the label fraction in some fatty acid species in GCN5-KD cells (Fig. 2D). Moreover, no changes were observed in any other metabolic pathways (Fig. EV2C), consistent with the previous finding (Fig. EV1B). These results argue that glucose-derived carbons are not associated with increased lipid accumulation in acetyltransferase-depleted cells. Further support-ing this conclusion, double-silencing of the acetyl-CoA synthetase ACLY together with each acetyltransferase did not diminish lipid droplet accumulation (Fig. EV2D–F). Overall, these data suggest that acetyltransferase-deficient hepatocytes do not use exogenous glucose-derived carbon for lipid synthesis.

## Hyperacetylated H4 is a carbon source for lipid synthesis upon acetyltransferase depletion

Since glucose-derived acetyl-CoA was not the source for lipid synthesis in acetyltransferase-deficient cells we subsequently asked whether histone-derived acetyl-CoA could be the source for lipid synthesis. When the histone acetylome was initially examined through MS mapping, the hyperacetylated form of histone H4 peptide H4K5K8K12K16-3ac showed a decreasing trend in all acetyltransferase-depleted cells, while the levels of the other acetylated peptides remained relatively unchanged (Figs. 3A and EV3A). Since this hyperacetylated histone peptide has been previously postulated to function as a carbon reservoir (Nirello

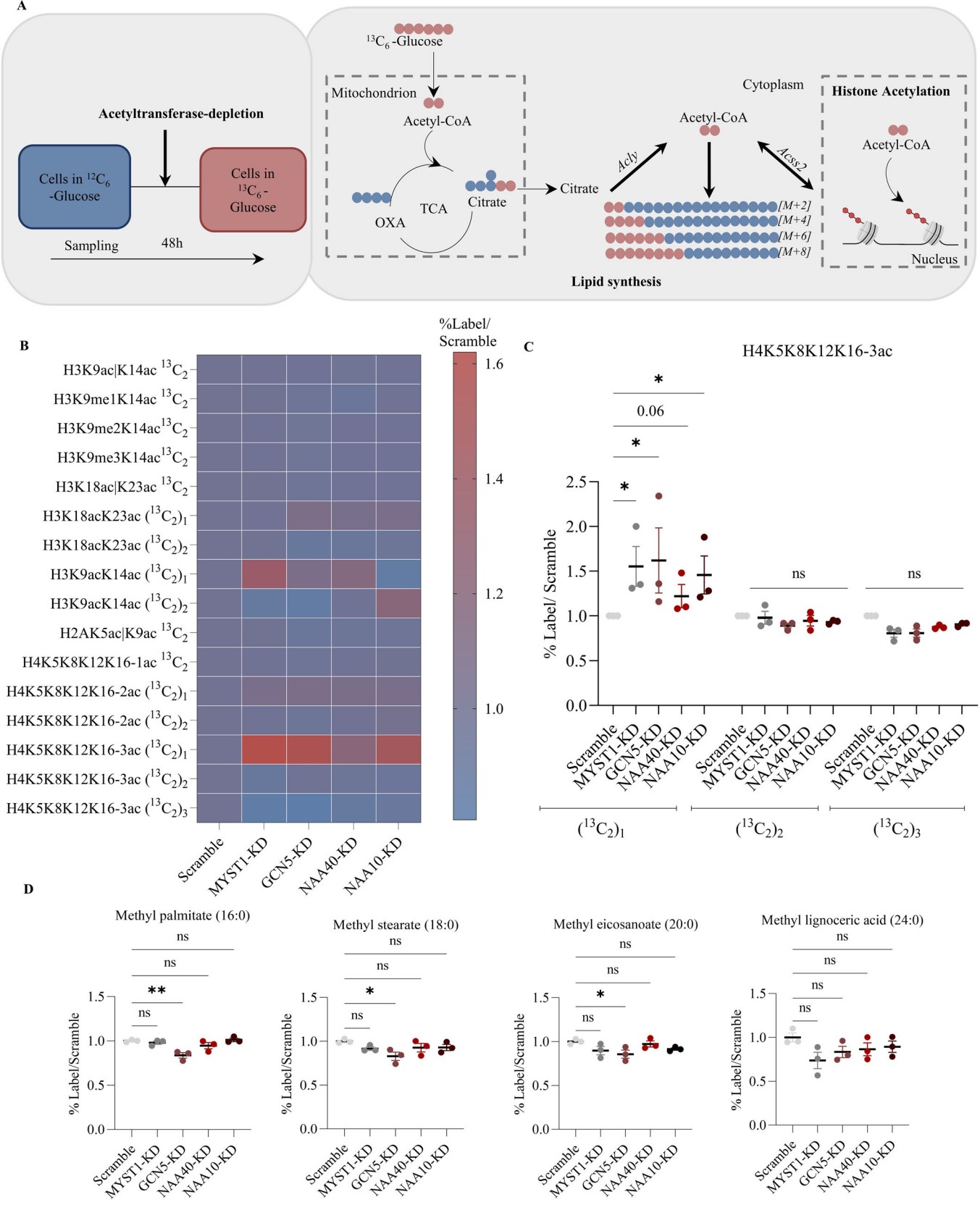

**Figure 2.  Lipid synthesis upon acetyltransferase depletion is not associated with glucose-derived acetyl-CoA in AML12 hepatocytes.**

(A) Schematic representation of experimental design (left) and the synthesis of cytosolic acetyl-CoA from exogenous glucose (right). (B) Heatmap of normalised levels (% label in KD/%label in Scramble) of $^{13}$C-labelled species in mono-, di- and tri-acetylated histone peptides in scramble, MYST1-KD, GCN5-KD, NAA40-KD and NAA10-KD cells. $n = 3$ biological replicates/group. (C) Normalised levels (%label in KD/%label in Scramble) of the tri-acetylated H4K5K8K12K16 peptide carrying one, two or three labelled acetyl-CoA moieties in scramble, MYST1-KD, GCN5-KD, NAA40-KD and NAA10-KD cells. $n = 3$ biological replicates/group. Statistical analysis was performed using two-way ANOVA with post hoc Tukey's multiple-comparisons test; *$P \leq 0.05$, ns non-significant. (D) Normalised levels (%label in KD/%label in Scramble) of indicated methyl fatty acids in scramble, MYST1-KD, GCN5-KD, NAA40-KD and NAA10-KD cells. $n = 3$ biological replicates/group. Statistical analysis was performed using one-way ANOVA with post hoc Dunnett's multiple-comparisons test; *$P \leq 0.05$, **$P \leq 0.01$, ns non-significant. Data information: (C, D) are presented as mean ± SEM. Source data are available online for this figure.

et al, 2022; McDonnell et al, 2016; Hamsanathan et al, 2022; Mendoza et al, 2022), the question arose as to whether carbons from this peptide can be used for lipogenesis. To investigate this, AML12 hepatocytes were first supplemented with $^{13}$C$_6$-Glucose for 7 days, which yielded a 77–85% incorporation of the $^{13}$C label in the histone fraction (Fig. EV3B), prior to acetyltransferase depletion. Having previously seen high lipid droplet formation in GCN5-KD and NAA40-KD hepatocytes (Fig. 1D–F), we proceeded with the depletion of these acetyltransferases in the labelled cells. Specifically, after 7-day $^{13}$C$_6$-Glucose labelling and following 12 h of siRNA induction, at which point the protein levels of both GCN5 and NAA40 were significantly diminished (Fig. EV3C), the medium was switched to unlabelled glucose ($^{12}$C$_6$-Glucose), in order to trace potential labelling derived from histones and not from exogenous glucose (Fig. 3B). While most histone peptides did not show changes between scramble and acetyltransferase-knockdowns (Fig. 3C–E and EV3D–I), the hyperacetylated form of histone H4 peptide H4K5K8K12K16-3ac was altered. In particular, the triply labelled version of the acetylated peptide significantly decreased at 18 h (corresponding to 6 h post media switch) in acetyltransferase-KD cells compared to control (Fig. 3F). Again, the tetra-acetylated form of the same peptide (4ac) was not detected in this analysis. Corresponding to the above finding, the percentage of $^{13}$C label in saturated methyl fatty acids (Fig. 3G–J) and in unsaturated methyl fatty acids (Fig. EV3J,K) increased significantly in acetyltransferase-KD cells compared to control at the same time point, without any changes in free fatty acids (Fig. EV3L–O) or any other metabolic pathway (Fig. EV3P). The isotopologue distribution analysis further validates our results, showing significant enrichment of labelled mass isotopologues of these methyl fatty acids in acetyltransferase-KD cells compared to scramble cells (Fig. 3K–N). The results of this isotope tracing analysis provide evidence that histone-derived carbon from hyperacetylated histone H4 contributes to lipid synthesis upon acetyltransferase depletion.

## Acetyltransferase-dependent lipid accumulation is associated with ACSS2-derived acetyl-CoA

To provide further evidence for histone-derived carbon as a significant source for lipid synthesis in acetyltransferase-deficient cells, HDACs were inhibited using sodium butyrate. HDACs generate acetate from histone deacetylation, which can then be converted into acetyl-CoA via ACSS2 (Mendoza et al, 2022). Therefore, we hypothesised that inhibition of HDACs could block the generation of acetate, fuelling histone acetylation rather than lipid synthesis in the acetyltransferase-depleted cells. Indeed, HDAC inhibition increased the levels of H4K8ac as expected

(Fig. EV4A,C) and concurrently blocked lipid droplet formation in acetyltransferase-KD hepatocytes (Fig. EV4B,C). Consistent with a role for deacetylation-derived acetate in lipid formation upon acetyltransferase-KD, depletion of either MYST1, GCN5, NAA40 or NAA10 significantly increased transcript (Fig. 4A) and protein levels (Fig. 4B) of ACSS2 compared to control cells, pointing towards a requirement for the expression of this acetyl-CoA synthetase. Supporting this notion, in NAA40-KD or GCN5-KD cells that respectively overexpress GCN5 or NAA40, ACSS2 is not induced (Fig. EV4D) and, as shown above, enhanced lipid synthesis in these cells is eliminated (Fig. 1H).

To conclusively examine the role of ACSS2 in this process, we depleted ACSS2 together with each acetyltransferase (Fig. EV4E) and found that lipid droplet formation and triglyceride content were completely abolished (Fig. 4C–E). Moreover, adhering to the same protocol of initially enriching histones with $^{13}$C for a duration of 7 days (Fig. 4F), and subsequent concurrent depletion of ACSS2 with either acetyltransferase GCN5 or NAA40 resulted in the sequestration of the label within the histone fraction. This sequestration was evidenced by an increase in the average of normalised levels of fully labelled $^{13}$C-histone peptides (Fig. 4G), which was primarily driven by an increase in H4K5K8K12K16-3ac and H3K9/14ac in cells subjected to double depletion (Fig. 4H). Importantly, the percentage of labelled fatty acids remained unchanged in cells lacking an acetyltransferase together with ACSS2 (Fig. 4I). Altogether, these findings demonstrate that hyperacetylated histone H4 is a source for HDAC-derived carbons, which through the presence of ACSS2, contribute towards lipid synthesis upon acetyltransferase depletion.

## HFHG-induced lipogenesis reduces hyperacetylated histone H4 and is alleviated by acetyltransferase overexpression

It was previously demonstrated that ACSS2 can significantly contribute to lipogenic acetyl-CoA independently of ACLY (Zhao et al, 2020). Since acetyltransferase depletion contributed to the accumulation of lipid droplets through the deacetylation of hyperacetylated histone H4 in an ACSS2-dependent manner, we next sought to examine whether the levels of hyperacetylated histone H4 are affected in diet-induced-lipogenesis. We first addressed this in vitro by supplementing hepatocytes with high fructose and glucose-(HFHG) as a tool to induce lipid droplet accumulation. We found that H4K5K8K12K16-3ac was among the most significantly decreased peptides in cells supplemented with HFHG media compared to control (Fig. 5A), and this corresponded with a significant increase in lipid droplets and triglycerides (Fig. 5B–D). Based on our findings above, which show that

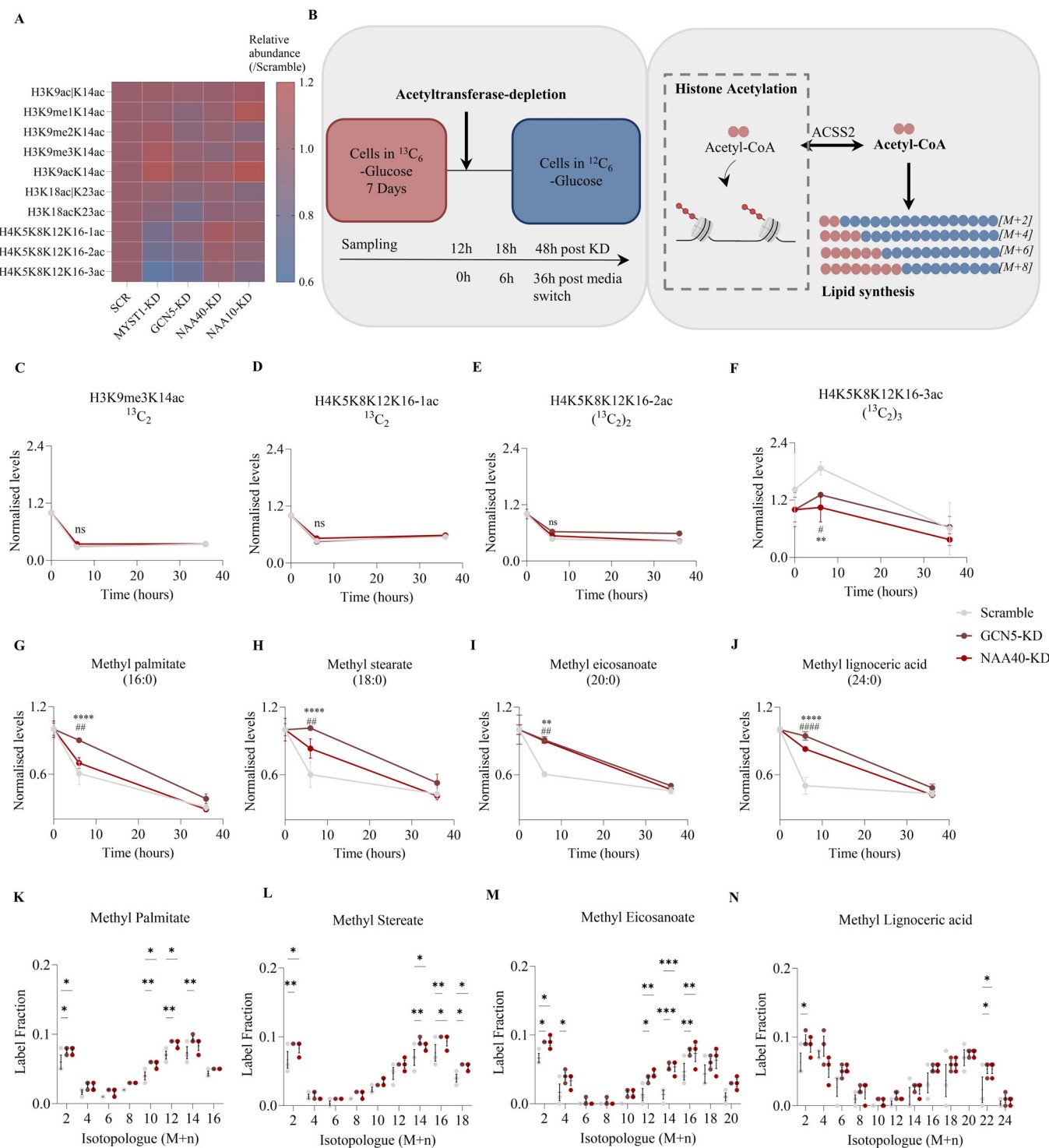

enhancement of overall acetyltransferase expression prevents lipid accumulation (Fig. 1H), we next assessed if increased acetyltransferase acetyl-CoA consumption can reverse high fructose/glucose-(HFHG)-induced lipid droplet accumulation (Liu et al, 2018). To address this, we co-overexpressed GCN5 and NAA40 (dO/E) in hepatocytes supplemented with HFHG (Fig. 5E) and found that lipid formation and triglyceride levels were decreased, as well as ACSS2 levels were reduced, in HFHG dO/E cells compared to non-

overexpressing HFHG-supplemented cells (Fig. 5B–F). Notably, the triply acetylated form of histone H4 tail was not reduced in acetyltransferase overexpressing cells supplemented with HFHG when compared to equivalent cells not supplemented with HFHG (Figs. 5G and EV5).

Subsequently, we repeated this investigation using an in vivo HFHG feeding study in mice for 6 weeks to induce liver lipid droplet accumulation (Fig. 6A). Mice supplied with HFHG water

**Figure 3.   Hyperacetylated H4 is a carbon source for lipid synthesis upon acetyltransferase depletion.**

(A) Heatmap of relative abundances (normalised to scramble) of histone acetylation marks measured by MS in scramble, MYST1-KD, GCN5-KD, NAA40-KD, and NAA10-KD cells. $n = 3$ biological replicates/group. (B) Schematic representation of experimental design (left) and the generation of acetyl-CoA from histone deacetylation (right). (C) Normalised levels (%label 6 h/%label 0 h) of mono-acetylated H3AK9me3K14 in scramble, GCN5-KD and NAA40-KD cells. $n = 3$ biological replicates/group. Statistical analysis was performed using two-way ANOVA with post hoc Tukey's multiple-comparisons test; ns non-significant. (D) Normalised levels (%label 6 h/%label 0 h) of H4K5K8K12K16-1ac carrying one labelled acetyl-CoA moiety in scramble, GCN5-KD, and NAA40-KD cells. $n = 3$ biological replicates/group. Statistical analysis was performed using two-way ANOVA with post hoc Tukey's multiple-comparisons test; ns non-significant. (E) Normalised levels (%label 6 h/%label 0 h) of H4K5K8K12K16-2ac carrying two labelled acetyl-CoA moieties in scramble, GCN5-KD and NAA40-KD cells. $n = 3$ biological replicates/group. Statistical analysis was performed using two-way ANOVA with post hoc Tukey's multiple-comparisons test; ns non-significant. (F) Normalised levels (%label 6 h/%label 0 h) of H4K5K8K12K16-3ac carrying three labelled acetyl-CoA moieties in scramble, GCN5-KD, and NAA40-KD cells. $n = 3$ biological replicates/group. Statistical analysis was performed using two-way ANOVA with post hoc Tukey's multiple-comparisons test; *$P \leq 0.05$, **$P \leq 0.01$. *for comparisons between scramble and GCN5-KD and # for scramble and NAAA40-KD. (G) Normalised levels (%label 6 h/%label 0 h) of methyl palmitate in scramble, GCN5-KD, and NAA40-KD cells. $n = 3$ biological replicates/group. Statistical analysis was performed using two-way ANOVA with post hoc Tukey's multiple-comparisons test; **$P \leq 0.01$, ****$P \leq 0.0001$. *For comparisons between scramble and GCN5-KD and # for scramble and NAAA40-KD. (H) Normalised levels (%label 6 h/%label 0 h) of methyl stearate in scramble, GCN5-KD, and NAA40-KD cells. $n = 3$ biological replicates/group. Statistical analysis was performed using two-way ANOVA with post hoc Tukey's multiple-comparisons test; **$P \leq 0.01$, ****$P \leq 0.0001$. *For comparisons between scramble and GCN5-KD and # for scramble and NAAA40-KD. (I) Normalised levels (%label 6 h/%label 0 h) of methyl eicosanoate in scramble, GCN5-KD and NAA40-KD cells. $n = 3$ biological replicates/group. Statistical analysis was performed using two-way ANOVA with post hoc Tukey's multiple-comparisons test; **$P \leq 0.01$. *For comparisons between scramble and GCN5-KD and # for scramble and NAAA40-KD. (J) Normalised levels (%label 6 h/%label 0 h) of methyl lignoceric acid in scramble, GCN5-KD, and NAA40-KD cells. $n = 3$ biological replicates/group. Statistical analysis was performed using two-way ANOVA with post hoc Tukey's multiple-comparisons test; ****$P \leq 0.0001$. *For comparisons between scramble and GCN5-KD and # for scramble and NAAA40-KD. (K) Mass isotopomer distribution of methyl palmitate at 6 h post media change in scramble, GCN5-KD, and NAA40-KD cells. $n = 3$ biological replicates/group. Statistical analysis was performed using two-way ANOVA with post hoc Tukey's multiple-comparisons test; *$P \leq 0.05$, **$P \leq 0.01$. (L) Mass isotopomer distribution of methyl stearate at 6 h post media change in scramble, GCN5-KD, and NAA40-KD cells. $n = 3$ biological replicates/group. Statistical analysis was performed using two-way ANOVA with post hoc Tukey's multiple-comparisons test; *$P \leq 0.05$, **$P \leq 0.01$. (M) Mass isotopomer distribution of methyl eicosanoate at 6 h post media change in scramble, GCN5-KD, and NAA40-KD cells. $n = 3$ biological replicates/group. Statistical analysis was performed using two-way ANOVA with post hoc Tukey's multiple-comparisons test; *$P \leq 0.05$, **$P \leq 0.01$, ***$P \leq 0.001$. (N) Mass isotopomer distribution of methyl lignoceric acid at 6 h post media change in scramble, GCN5-KD, and NAA40-KD cells. $n = 3$ biological replicates/group. Statistical analysis was performed using two-way ANOVA with post hoc Tukey's multiple-comparisons test; *$P \leq 0.05$. Data information: (C–N) are presented as mean ± SEM. Source data are available online for this figure.

demonstrated greater body weight gain by week 6 compared to control mice on tap water (~4% increase in body weight; Fig. 6B). In agreement with the in vitro results, the livers of mice on HFHG water showed a significant decrease in H4K5K8K12K16-3ac, which was concomitant to lipid droplet and triglyceride accumulation compared to livers from mice consuming tap water (Fig. 6C–F, compare HFHG EV vs tap-water EV). To determine whether enhanced acetyltransferase expression has an effect on HFHG-induced lipogenesis in vivo, mice from each group received a single intraperitoneal (IP) injection of plasmids, which are absorbed from the peritoneal cavity to the portal vein and end up in the liver (Al Shoyaib et al, 2020), thus overexpressing both GCN5 and NAA40 (dO/E) or a cocktail of their corresponding empty vectors (EV), upon the end of week 6 (Fig. 6A). Following co-transfection with IP injections, mice on HFHG water which also received both GCN5 and NAA40 overexpressing plasmids, displayed induced expression of these acetyltransferases in their livers and lost significant weight (decreased their body weight by 4%) returning to the control mouse weight levels (Fig. 6B). Remarkably, in mouse livers overexpressing GCN5 and NAA40 (Fig. 6G), lipid droplet formation and triglyceride accumulation was prevented and the levels of ACSS2 were reduced in HFHG dO/E livers compared to non-overexpressing HFHG livers (Fig. 6D–G). Moreover, H4K5K8K12K16-3ac levels were not reduced in acetyltransferase overexpressing livers supplemented with HFHG when compared to overexpressing livers that were not supplemented with HFHG (Fig. 6H). Overall, the results of both the in vitro and in vivo studies suggest that histone-derived carbon, from hyperacetylated histone H4, could be a source for lipid synthesis in diet-induced lipogenesis. In addition, these results demonstrate that enhanced acetyltransferase expression could alleviate diet-induced fat accumulation.

## Discussion

Accumulating evidence in recent years highlighted the strong regulatory interactions between epigenetic modifications and metabolic states, mainly through transcriptional control of key metabolic genes (McDonnell et al, 2016; Gao et al, 2016; Li et al, 2017). In addition, the potential role of histones as metabolite reservoirs has been postulated due to the high abundance of histone proteins in cells, which can potentially store substantial amounts of metabolic intermediates like acetyl-CoA (Ye and Tu, 2018). However, evidence supporting the contribution of histones to metabolic rewiring has not yet been reported (Ye and Tu, 2018; Wong et al, 2017; Katada et al, 2012; Lozoya et al, 2018). In the present study, we show for the first time the potential of hyperacetylated histone H4 to contribute carbons toward lipid synthesis under conditions that induce lipogenesis in hepatocytes.

In particular, using stable isotope tracing, we demonstrate within cells that this increased lipid formation potentially arises from the deacetylation of the tri-acetylated form of the histone H4 tail, contributing to the generation of acetyl-CoA, through ACSS2. In our experiments, we were not able to quantify the differentially labelled forms of tetra-acetylated (4ac) histone H4, but it is plausible that this hyperacetylated form may also act as a reservoir of acetylation. Corroborating our isotope tracing results, further findings in hepatocytes and mouse liver show the ability of increased acetyltransferase expression to reverse diet-induced lipid accumulation and prevent the decrease in the levels of the hyperacetylated H4. These observations expand the link between epigenetics and metabolism, whereby chemical groups from deacetylation of hyperacetylated histone peptides can directly lead to the generation of lipogenic acetyl-CoA, potentially contributing to the development of fatty liver (Fig. 7).

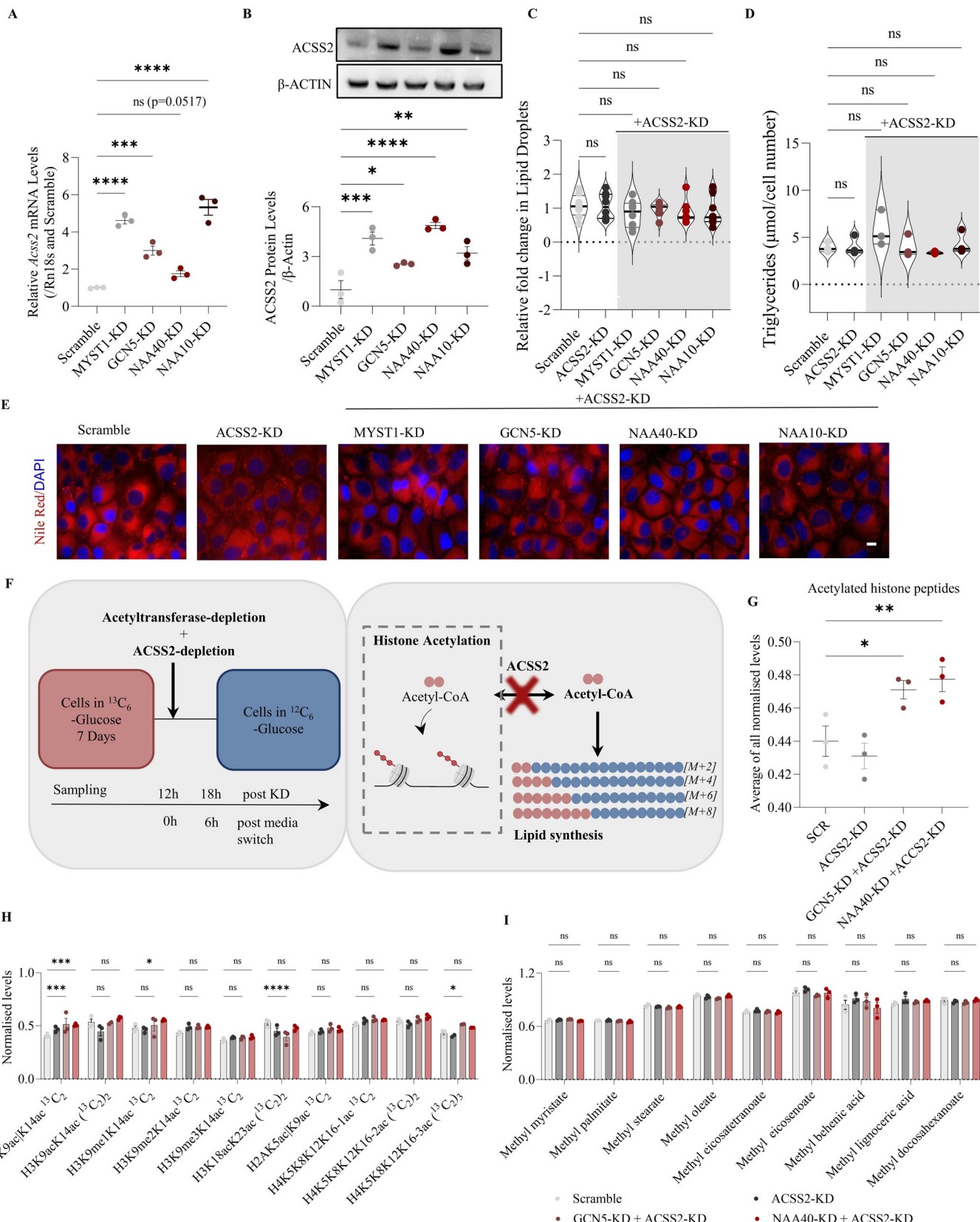

**Figure 4. ACSS2 drives HDAC-dependent lipid droplet formation upon acetyltransferase depletion in AML12 hepatocytes.**

(A) RT-qPCR analysis of *Acss2* mRNA levels in scramble, MYST1-KD, GCN5-KD, NAA40-KD cells, and NAA10-KD 48 h after siRNA treatment. $n = 3$ biological replicates/ group. Statistical analysis was performed using one-way ANOVA with post hoc Dunnett's multiple-comparisons test; ***$P \leq 0.001$, ****$P \leq 0.0001$, ns non-significant. (B) Representative immunoblot (top) and quantification (bottom) of ACSS2 in scramble, MYST1-KD, GCN5-KD, NAA40-KD cells, and NAA10-KD 48 h after siRNA treatment. $n = 3$ biological replicates/group. Statistical analysis was performed using one-way ANOVA with post hoc Dunnett's multiple-comparisons test; *$P \leq 0.05$, **$P \leq 0.01$, ***$P \leq 0.001$, ****$P \leq 0.0001$. (C) Quantification of relative lipid droplets in scramble and the indicated double-KD cells after 48 h of siRNA treatment. $n = 6$–8 biological replicates/group. Statistical analysis was performed using one-way ANOVA with post hoc Dunnett's multiple-comparisons test; ns non-significant. (D) Triglyceride levels in scramble and the indicated double-KD cells after 48 h of siRNA treatment. $N = 3$ biological replicates/group. Statistical analysis was performed using one-way ANOVA with post hoc Dunnett's multiple-comparisons test; ns non-significant. (E) Representative images of lipid droplets by Nile red (red) and nuclei by DAPI (blue) in scramble and the indicated double-KD cells after 48 h of siRNA treatment. Scale bar = 25 μm. $n = 6$–8 biological replicates/group. (F) Schematic representation of experimental design (left) and the generation of acetyl-CoA from histone deacetylation, through the activity of ACSS2 (right). (G) Average of normalised levels (%label 6 h/%label 0 h) of the detected fully labelled acetylated histone peptides in scramble, ACSS2-KD, and double-KD cells. $n = 3$ biological replicates/group. Statistical analysis was performed using one-way ANOVA with post hoc Dunnett's multiple-comparisons test; *$P \leq 0.05$, **$P \leq 0.01$. (H) Normalised levels (%label 6 h/%label 0 h) of indicated histone peptides in scramble, ACSS2-KD, and double-KD cells. $n = 3$ biological replicates/group. Statistical analysis was performed using two-way ANOVA with post hoc Tukey's multiple-comparisons test; *$P \leq 0.05$, ***$P \leq 0.001$, ****$P \leq 0.0001$, ns non-significant. (I) Normalised levels (%label 6 h/%label 0 h) of indicated methyl fatty acids in scramble, ACSS2-KD, and double-KD cells. $n = 3$ biological replicates/group. Statistical analysis was performed using two-way ANOVA with post hoc Tukey's multiple-comparisons test; ns non-significant. Data information: (A, B, G–I) are presented as mean ± SEM; (C, D) are presented as violin plots. Source data are available online for this figure.

Our model demonstrates a dynamic relationship between the relative overall expression of acetyltransferases and HDACs acting on hyperacetylated histone reservoirs to modulate lipid synthesis. We provide various evidence that this relationship is mediated by the common activity of acetyltransferases to consume acetyl-CoA rather than their target specificity and specific transcriptional outcomes, whereby acetyl groups from hyperacetylated histone reservoirs preferentially contribute to lipids rather than overall histone or protein acetylation in the absence of acetyltransferases. Firstly, the depletion of four distinct acetyltransferases (including both NATs and KATs), with different histone and non-histone target specificities, results in increased lipid synthesis in hepatocytes (Fig. 1A–F). More specifically, the use of the protein acetyltransferase NAA10 in this study indicates the potential of other acetyl-CoA consumers, beyond histone acetyltransferases, to directly affect histone reservoirs and lipid synthesis without altering the chromatin state and thereby transcription of lipid synthesis genes, further supporting our model of overall reduced acetylation activity contributing to lipid synthesis through histone reservoirs (Fig. 7). Secondly, HDAC inhibition in all four acetyltransferase-depleted cells alleviates induction of lipid droplet accumulation (Fig. 4). Thirdly, overexpressing one of these acetyltransferases in the absence of another (i.e., GCN5 in NAA40-KD or NAA40 in GCN5-KD cells) rescues the acetyltransferase-depletion effect by preventing the accumulation of lipid droplets (Fig. 1G,H), suggesting again that overall acetyl-CoA consumption is probably what associates to lipid synthesis and not specific transcriptional programmes. Furthermore, together with our previous finding (Charidemou et al, 2022), we show that lipid synthesis upon NAA40 depletion is induced faster than any transcriptional changes of lipid synthesis-related genes (Fig. EV1D–F), further supporting the scenario that lipid induction upon acetyltransferase depletion occurs independently of transcriptional effects but rather because of reduced overall acetyl-CoA consumption. In agreement with this, it has been recently demonstrated that the activity of histone methyltransferases impacts cellular metabolism in human cells independently of transcriptional regulation (Perez and Sarkies, 2023). Interestingly, the expression of ACCS2 seems to be changing in all acetyltransferase-depleted cells, suggesting a generic response to acetate availability possibly arising from the perturbation in acetylation/deacetylation equilibrium. This observation is supported by the fact that ACSS2 expression is altered under different metabolic stressors to promote acetate utilisation (Schug et al, 2015), however, the precise mechanism inducing its transcription remains unknown.

In line with the scenario of histones serving as carbon reservoirs, other studies previously hypothesised that hyperacetylated histone peptides carrying acetyl marks at multiple sites may have a potential to act as reservoirs (Nirello et al, 2022; McDonnell et al, 2016; Hamsanathan et al, 2022; Mendoza et al, 2022). Consistent with this notion, our stable isotope tracer study analytically demonstrates that removal of labelled carbons from hyperacetylated lysine residues of histone peptide H4 associates with increased labelling in fatty acids upon acetyltransferase depletion (Fig. 3). We did not observe any changes in mono- or di-acetylated H4 peptides, implying that these could present acetylated states that associate mainly with transcriptional effects. By using an MS bottom-up approach, we were able to measure the levels of histone H3 doubly acetylated peptides H3K9acK14ac and H3K18acK23ac, and found no differences, suggesting these histone H3 peptides do not contribute, or contribute to a lesser extent, carbons to lipid synthesis. However, our analysis does not exclude the possibility of hyperacetylated H3 (H3K9acK14acK18acK23ac) serving as a carbon donor, since we were not able to detect this peptide with our bottom-up MS approach and it was previously proposed to serve as an acetate reservoir (Hamsanathan et al, 2022). In order for histones to act as reservoirs, there is a requirement for HDACs to derive acetate from them and then ACCS2 to convert this acetate to acetyl-CoA (Mendoza et al, 2022). Accordingly, our findings show that both HDAC activity and ACSS2 expression are required for lipid induction in acetyltransferase-depleted cells. This is again consistent with previous studies which demonstrate that HDAC activity and ACSS2 are responsible for direct acetate transfer between histone lysine residues, as well as suggest that HDAC inhibition may affect the release of free acetate that is recaptured to generate acetyl-CoA (Hsieh et al, 2022). Finally, our model of hyperacetylated histones contributing carbons directly to lipid synthesis (Fig. 7), is coherent with recent work from Soaita et al, (2023), which demonstrates the contribution of deacetylation-derived carbons towards cellular acetyl-CoA and downstream metabolites, such as the lipid intermediate (iso)butyryl-CoA. The authors also highlight that this carbon contribution might even be more substantial when anabolism is activated (Soaita et al, 2023) which is the case

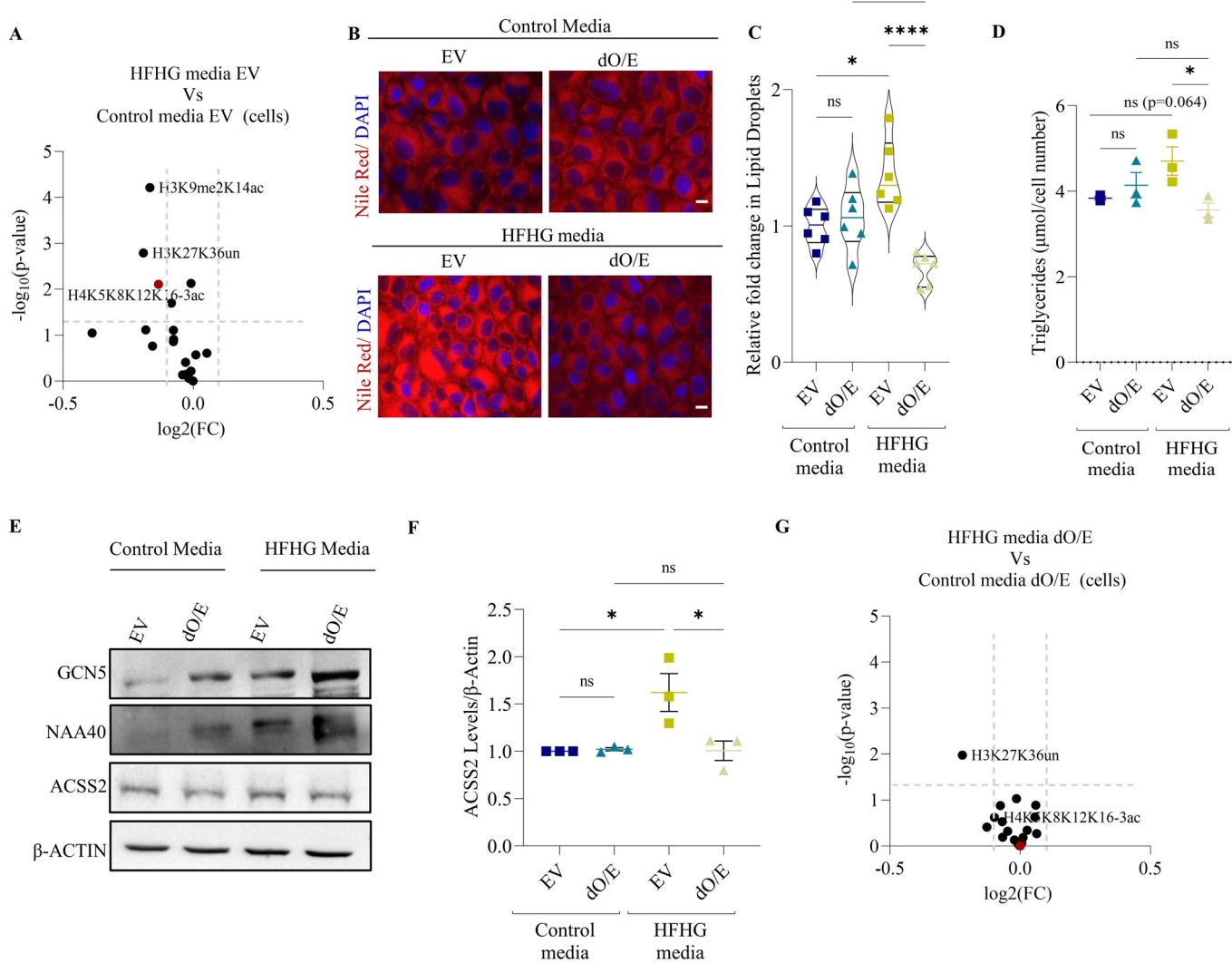

**Figure 5. Overexpression of acetyltransferases reverses HFHG-induced fat accumulation in vitro.**

(A) Volcano plot comparing histone acetylation marks, measured by MS, in cells with high fructose/high glucose (HFHG) media vs. control Media carrying EVs. $n = 3$ biological replicates/group. Statistical analysis was performed using a two-tailed T test. (B) Representative images of lipid droplets by Nile red (red) and nuclei by DAPI (blue) in control media, and high fructose/high glucose (HFHG) media, 48 h after co-overexpression of GCN5 and NAA40 (dO/E) or corresponding empty vectors (EV) in AML12 hepatocytes; Scale bar = 25 µm. $n = 6$ biological replicates/group. (C) Quantification of relative lipid droplets in control media, and high fructose/high glucose (HFHG) media, 48 h after co-overexpression of GCN5 and NAA40 (dO/E) or corresponding empty vectors (EV) in AML12 hepatocytes. $n = 6$ biological replicates/group. Statistical analysis was performed using one-way ANOVA with post hoc Dunnett's multiple-comparisons test; *$P \le 0.05$, ****$P \le 0.0001$, ns non-significant. (D) Triglyceride levels in control media, and high fructose/high glucose (HFHG) media, 48 h after co-overexpression of GCN5 and NAA40 or corresponding empty vectors (EV) in AML12 hepatocytes. $n = 3$ biological replicates/group. Statistical analysis was performed using one-way ANOVA with post hoc Dunnett's multiple-comparisons test; *$P \le 0.05$, ns non-significant. (E) Representative immunoblots of GCN5, NAA40, ACSS2 and β-actin in control media, and high fructose/high glucose (HFHG) media, 48 h after co-overexpression of GCN5 and NAA40 (dO/E) or corresponding empty vectors (EV) in AML12 hepatocytes. $n = 3$ biological replicates/group. (F) Quantification of ACCS2 in control media and high fructose/high glucose (HFHG), 48 h after co-overexpression of GCN5 and NAA40 (dO/E) or corresponding empty vectors (EV) in AML12 hepatocytes. $n = 3$ biological replicates/group. Statistical analysis was performed using one-way ANOVA with post hoc Dunnett's multiple-comparisons test; *$P \le 0.05$, ns non-significant. (G) Volcano plot comparing histone acetylation marks, measured by MS, in cells with HFHG media Vs Control media carrying both GCN5 and NAA40 plasmids (dO/E) in AML12 hepatocytes. $n = 3$ biological replicates/group. Statistical analysis was performed using a two-tailed T test. Data information: Log2(FC) threshold for (A, G) based on variance=0.018. (C) is presented as a violin plot; (D, F) are presented as mean ± SEM. Source data are available online for this figure.

during acetyltransferase depletion (Charidemou et al, 2022). Although our evidence supports histones as a source of carbon, we cannot exclude the potential existence of additional compensatory fluxes aimed at reinstating metabolic homoeostasis. Further stable isotope analysis may be necessary to comprehensively explore and elucidate other potential compensatory metabolic fluxes.

The generation of lipogenic acetyl-CoA, through the activity of ACCS2 on acetate, has been found to be responsible for fructose-induced fatty liver through increased lipid synthesis rates (Zhao et al, 2020; Schug et al, 2015). In fact, the increasing amounts of fructose in the diet in the form of sugary beverages or processed foods leads to hepatic fat accumulation, inflammation and possibly

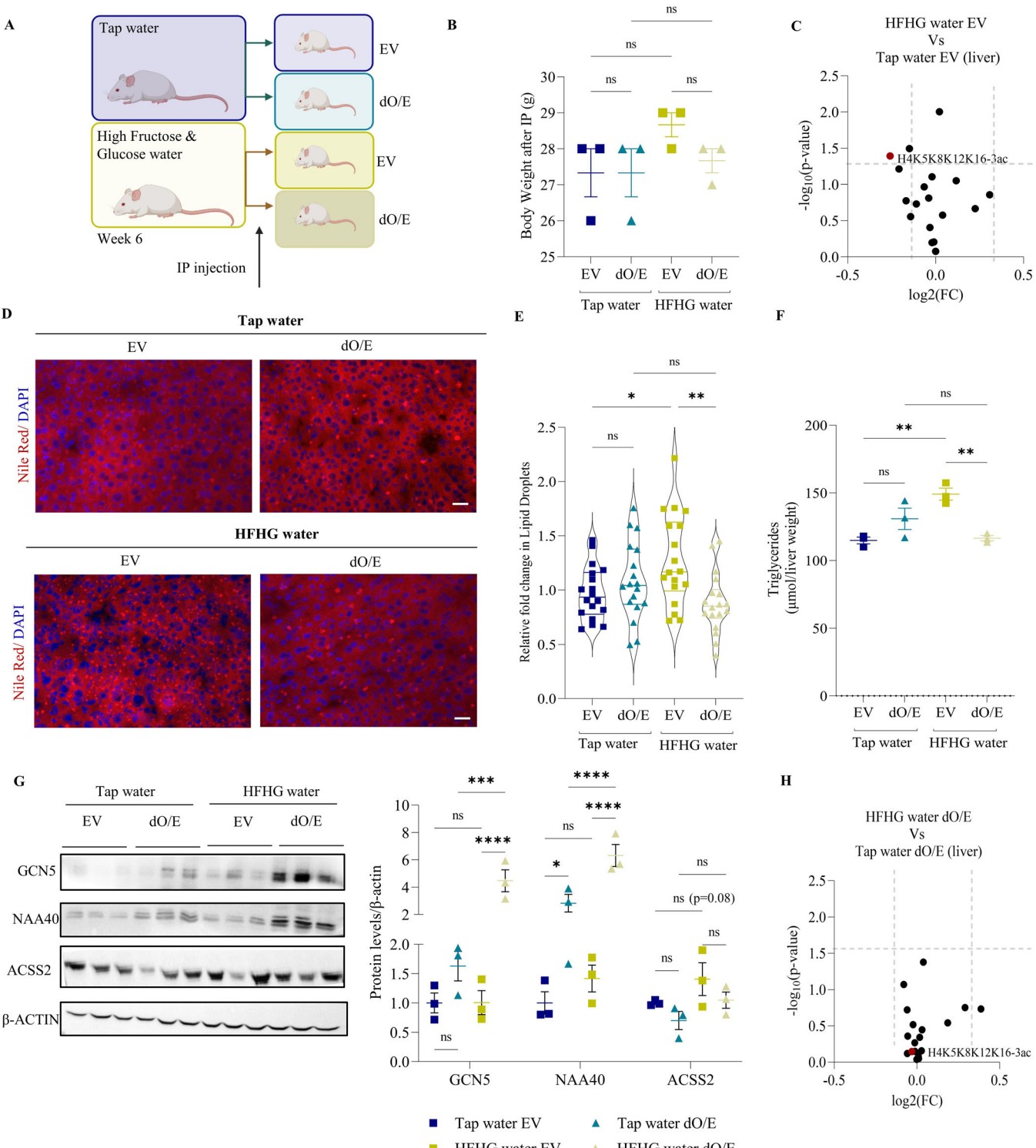

fibrosis, contributing to the development and severity of non-alcoholic fatty liver disease (NAFLD) (Vos and Lavine, 2013). Our results show a potential role for histone acetylation reservoirs in the generation of lipogenic acetyl-CoA in diet-induced fat accumulation (Fig. 7). Hence, these findings raise the possibility that histone acetylation reservoirs are implicated in diet-triggered NAFLD, and this would be interesting to address in future investigations.

Overall, this study proposes a novel role of histone acetylation in the metabolism-epigenetics nexus, whereby hyperacetylated histone H4 represents a metabolic reservoir and the relative activity between acetyltransferases and HDACs can impact this reservoir, thereby determining lipogenic acetyl-CoA pools (Fig. 7). This emerging concept may apply to other histone modifications, such as acylations, and their associated metabolites, as well as have

**Figure 6.  Overexpression of acetyltransferases reverses HFHG-induced fat accumulation in vivo.**

(A) Schematic representation of in vivo experimental design. (B) Body weight (grams) of mice supplied with tap water or HFHG water 48 h after IP injections with both GCNA and NAA40 (dO/E) or their corresponding empty vectors (EV). $n = 3$ biological replicates/group. Statistical analysis was performed using one-way ANOVA with post hoc Dunnett's multiple-comparisons test; ns non-significant. (C) Volcano plot comparing histone acetylation marks, measured by MS, in livers from mice supplied with high fructose/high glucose (HFHG) water vs. tap water carrying EVs. $n = 3$ biological replicates/group. Statistical analysis was performed by a two-tailed $T$ test. (D) Representative images of lipid droplets by Nile red (red) and nuclei by DAPI (blue) of mouse liver sections from each treatment group. Scale bar = 25 μm. $n = 3$ biological replicates/group (9 technical replicates). (E) Quantification of relative lipid droplets of mouse liver sections from each treatment group. $n = 3$ biological replicates/group (9 technical replicates). Statistical analysis was performed using one-way ANOVA with post hoc Dunnett's multiple-comparisons test; *$P ≤ 0.05$, **$P ≤ 0.01$, ns non-significant. (F) Triglyceride quantification of mouse liver from each treatment group. $n = 3$ biological replicates/group. Statistical analysis was performed using one-way ANOVA with post hoc Dunnett's multiple-comparisons test; **$P ≤ 0.01$, ns non-significant. (G) Representative immunoblots (left) and quantification (right) of GCN5, NAA40, ACSS2 and β-actin in tap water and high fructose/high glucose (HFHG) conditions, 48 h after co-overexpression of GCN5 and NAA40 (dO/E) or corresponding empty vectors (EV). $n = 3$ biological replicates/group. Statistical analysis was performed using two-way ANOVA with post hoc Tukey's multiple-comparisons test; *$P ≤ 0.05$, ****$P ≤ 0.0001$, ns non-significant. (H) Volcano plot comparing histone acetylation marks in livers of mice supplied with HFHG water vs. tap water carrying both GCN5 and NAA40 plasmids (dO/E). $n = 3$ biological replicates/group. Statistical analysis was performed using a two-tailed $T$ test. Data information: Log2(FC) threshold for (C, H) based on variance = 0.02. (B, F–G) Are presented as mean ± SEM; (E) is presented as a violin plot. Source data are available online for this figure.

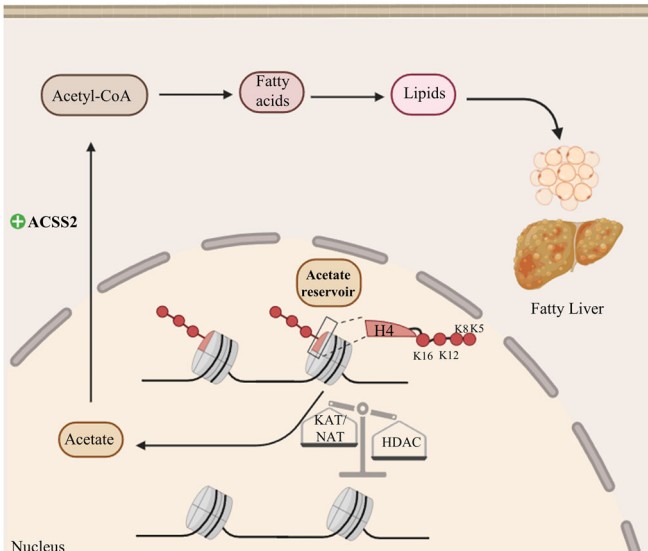

**Figure 7.  Model of hyperacetylated H4 reservoir contributing carbon to lipids.**

Based on the overall activity between acetyltransferases and HDACs, carbons derived from deacetylation of hyperacetylated histone H4 can contribute to lipid synthesis in an ACSS2-dependent manner.

important implications in diseases with increased rates of lipid synthesis like NAFLD (Kumashiro et al, 2011).

# Methods

## Cell culture

AML12 cells were purchased from American Type Culture Collection (ATCC) and cultured in 1:1 (v/v) Dulbecco's modified Eagle's medium and Ham's F12 medium (Thermo), supplemented with 10% foetal bovine serum (FBS), 1% penicillin/streptomycin (100 units/mL and 100 μg/mL, respectively), 1% insulin–transferrin–selenium (ITS; 10 mg/L, 5.5 mg/L and 6.7 μg/L, respectively) and dexamethasone (100 μmol/L) at 37 °C in 5% $CO_2$. Cells were plated at a density of 50,000 cells/well in collagen-I coated 12-well plates (cat. # 7340295, Corning) and grown to confluence in a maintenance medium. For

high fructose/glucose treatments, cells were supplemented with 25 mM of each L-fructose and L-glucose.

## siRNA-mediated knockdown

*Myst-1, Gcn5, Naa40, Naa10, Acly* and *Acss2* were silenced by siRNA transfection or each HAT in combination with *Acly* or *Accs2* were co-silenced in AML12 cells. In brief, cells were plated at a density of 100,000 cells/well and on day 2, cells were transfected with 25 nM siRNA specific for *Myst-1* (QIAgen; cat. #GS67773) or *Gcn5* (QIAgen; cat. # GS14534) or *Naa40* (QIAgen; cat. # GS70999) or *Naa10* (QIAgen; cat. # GS56292) or *Acly* (QIAgen; cat. # GS104112) or *Acss2* (Qiagen; cat. # GS60525) or negative control (Qiagen; cat. #1027281) using DharmaFECT 1 Transfection Reagent (Horizon discovery) according to the manufacturer's instructions. Cells were harvested at 6, 12, 24, 48 h post transfection.

## Transient overexpression

Mouse *Gcn5* and human *NAA40* plasmids were overexpressed in AML12 cells. In brief, cells were plated at a density of 100,000 cells/well and on day 2, cells were transfected with 1 μg plasmid DNA carrying mouse Gcn5 (pMSCVpuro-Gcn5, addgene cat. # 63706) or its corresponding empty vector (EV; pMSCVpuro-mApple, addgene cat. # 96934) or 1 μg plasmid DNA carrying full-length human NAA40 cDNA (pLenti/V5-NAA40) or an Empty vector as a non-coding control (EV; pLenti/V5-empty vector) using FuGENE HD (Promega Corporation) according to the manufacturer's instructions. For double overexpression experiments, 1 μg of each pMSCVpuro-Gcn5 and pLenti/V5-NAA40 or their corresponding empty vectors were used for co-transfection. Cells were harvested 48 h post transfection.

## Analysis of intact lipids using high-resolution mass spectrometry

Metabolites and lipids were extracted from cells using a modified method of Folch and colleagues (Folch et al, 1957). Briefly, pelleted cells were homogenised in chloroform/methanol (2:1, v/v, 750 μL); including a mixture of deuterated internal standards. Samples were sonicated for 15 min and deionised water was added (300 μL).

The organic (upper layer) and aqueous (lower layer) phases were separated following centrifugation at 13,000×*g* for 20 min. The organic phase extracts containing lipids were dried under a stream of nitrogen gas.

An internal standard mix was incorporated to the organic fraction [N-palmitoyl-d$_{31}$-D-erythro-sphingosine (16:0-d$_{31}$ Ceramide), pentadecanoic-d$_{29}$ acid (15:0-d$_{29}$ FFA), heptadecanoic-d$_{33}$ acid (17:0-d$_{33}$ FFA), eicosanoic-d$_{39}$ acid (20:0-d$_{39}$ FFA), 1-palmitoyl(D$_{31}$)-2-oleyl-sn-glycero-3-phosphatidylcholine (16:0-d$_{31}$-18:1 PC), 1-palmitoyl(d$_{31}$)-2-oleyl-sn-glycero-3-phosphoethanolamine (16:0-d$_{31}$-18:1 PE), 1-palmitoyl-d$_{31}$-2-oleoyl-sn-glycero-3-[phospho-rac-(1-glycerol)] (16:0-d$_{31}$-18:1 PG), N-palmitoyl(d$_{31}$)-d-erythro-sphingosylphosphorylcholine (16:0-d$_{31}$ SM), glyceryl-tri(pentadecanoate-d$_{29}$) (45:0-d$_{87}$ TAG), glyceryl-tri(hexadecanoate-d$_{31}$) (48:0-d$_{93}$ TAG) (Avanti Polar Lipids Inc, USA)] and dried down under nitrogen. The organic fraction was reconstituted in 100 μL chloroform/methanol (1:1, v/v), and 10 μL of the resulting solution was added to 90 μL isopropanol (IPA)/acetonitrile (ACN)/water (2:1:1, v/v). Analysis of the fractions was performed using an LTQ Orbitrap Elite Mass Spectrometer (Thermo Scientific, Hemel Hempstead, UK).

In positive mode, 5 μL of the sample were injected onto a C18 CSH column, 1.7 μM pore size, 2.1 mm × 50 mm (cat # 186005296, Waters Ltd, Manchester, UK), which was held at 55 °C in a Dionex Ultimate 3000 ultra-high performance liquid chromatography system (UHPLC; Thermo Scientific). A gradient (flow rate 0.5 mL/min) of mobile phase A (ACN/water 60:40, 10 mmol/L ammonium formate) and B (LC–MS-grade ACN/IPA 10:90, 10 mmol/L ammonium formate) was used. In negative ion mode, 10 μL of the sample was injected and 10 mmol/L ammonium acetate was used as the additive to aid ionisation.

In both positive and negative ion mode, the gradient began at 40% B, increased to 43% B at 0.8 min, 50% B at 0.9 min, 54% B at 4.8 min, 70% B at 4.9 min, 81% B at 5.8 min, peaked at 99% B at 8 min for 0.5 min, and subsequently returned to the starting conditions for another 1.5 min to re-equilibrate the column. The UHPLC was coupled to an electrospray ionisation (ESI) source which ionised the analytes before entering the mass spectrometer. Data were collected in positive ion mode with a mass range of 110–2000 *m/z*. Default instrument-generated optimisation parameters were used.

The spectra files were converted to mzML format and features were picked using xcms (Smith et al, 2006), after retention time alignment. Lipid identification was performed using an in-house R script. Peak areas of each metabolite were normalised to the appropriate internal standard.

## $^{13}C_6$-glucose labelling procedure in AML12 hepatocytes

For the first labelling experiment, cells were supplemented with $^{13}C_6$-glucose (7.60 mmol/L, *n* = 3) upon siRNA induction of MYST1 or GCN5 or NAA40 or NAA10 and were harvested at 48 h.

For the second labelling experiment, cells were supplemented with $^{13}C_6$-glucose (7.60 mmol/L, *n* = 3) for 7 days. siRNA knockdown of GCN5 and NAA40 was induced, and 12 h post transfection, the media was switched back to $^{12}C_6$-glucose. Cells were harvested at 0 h (12 h post siRNA), 6 h (18 h post siRNA) and 36 h (48 h post siRNA) post media switch. For both experiments, each cell pellet ($8 \times 10^6$ cells) was split in two, with one half used for further MS-based epigenetic mapping and the other for metabolomic analyses.

For the third labelling experiment, cells were supplemented with $^{13}C_6$-glucose (7.60 mmol/L, *n* = 3) for 7 days. siRNA knockdown of GCN5 or NAA40 in combination with ACSS2 was induced, and 12 h post transfection, the media was switched back to $^{12}C_6$-glucose. Cells were harvested at 0 h (12 h post siRNA) and 6 h (18 h post siRNA) post media switch. For both experiments, each cell pellet ($8 \times 10^6$ cells) was split in two, with one half used for further MS-based epigenetic mapping and the other for metabolomic analyses.

## Mass spectrometry-based histone post-translation modification analysis

Approximately 3–4 μg of histone octamer were separated on a 17% SDS-PAGE gel. Histone bands were excised, chemically acylated with propionic anhydride, and in-gel digested with trypsin, followed by peptide N-terminal derivatisation with phenyl isocyanate (PIC) (Noberini et al, 2021). Peptide mixtures were separated by reversed-phase chromatography on an EASY-Spray column (Thermo Fisher Scientific), 25 cm long (inner diameter 75 μm, PepMap C18, 2 μm particle size), which was connected online to a Q Exactive Plus or HF instrument (Thermo Fisher Scientific) through an EASY-Spray™ Ion Source (Thermo Fisher Scientific), as described (Noberini et al, 2021). The acquired RAW data were analysed using Epiprofile 2.0 (Yuan et al, 2018), selecting the "histone_C13" option, followed by manual validation (Noberini et al, 2021). For each acetylated peptide, a % relative abundance (%RA) was estimated by dividing the area under the curve (AUC) of $^{13}C_2$-labelled peptides with the sum of the areas corresponding to all the observed labelled/unlabelled forms of that peptide and multiplying by 100. For the analysis of unlabelled samples, histones were mixed with an equal amount of heavy-isotope labelled histones, which were used as an internal standard (PMID: 36370272) prior to loading on the gel. The SILAC option was selected in Epiprofile 2.0, and %RA values were calculated for the sample (light channel - L) and the internal standard (heavy channel—H). Light/Heavy (L/H) ratios of %RA were then calculated. %RA or L/H ratios are reported in https://doi.org/10.6084/m9.figshare.23257160. The MS-MS spectra for the differentially acetylated forms of the histone H4 tail (peptide H4K5K8K12K16) are shown in Fig. EV5.

## Mass spectrometry-based methyl fatty acid (lipid-bound) and free fatty acid analysis

To obtain separated organic and aqueous fractions, extracts were dried using an Eppendorf Vacufuge concentrator, and a dual-phase extraction was performed by adding 300 μl of CHCl$_3$/MeOH (2:1) and vortexing for 30 s. After adding 300 μl of water and centrifugation (13,000 rpm, 10 min, room temperature), the top aqueous layer was transferred into an inactivated glass vial and dried before being stored at −80 °C. The lower organic layers were placed into glass vials and dried overnight before being stored at −80 °C.

For transmethylation of the lipid-bound fatty acids, the dried fraction was reconstituted in 300 μL of methanol/toluene solution (1:1 ratio), treated with 200 μL of 0.5 M sodium methoxide and incubated for 1 h at room temperature. The reaction was stopped by adding 500 μL of 1 M NaCl and 25 μL of concentrated HCl.

The resulting samples were extracted by using 500 µL of hexane and organic layers were dried in the fume cupboard under N₂. Free fatty acids were then derivatized using 40 µL acetonitrile and 40 µL of N-(tert-butyldimethylsilyl)-N-methyl trifluoro-acetamide (MBTSFA; Thermo Fisher Scientific, Waltham, MA, USA) and incubated at 70 °C for 1 h. Samples were finally centrifuged at 2000 rpm for 5 min prior to transferring them into a clean vial insert for GC-MS analysis.

Aqueous fractions were derivatized by a two-step methoximation-silylation derivatization procedure. The dried samples were first methoximated using 20 µl of 20 mg/ml methoxyamine hydrochloride in anhydrous pyridine at 37 °C for 90 min. For unlabelled samples, this was followed by silylation with 80 µl of N-methyl-N-(trimethylsilyl) tri-fluoroacetamide (MSTFA) at 37 °C for 30 min. Labelled samples were derivatized by adding 80 µl of N-(tert-butyldimethylsilyl)-N-methyltrifluoroacetamide (MBTSFA [Thermo Scientific]). After being vortexed, samples were placed in a heater block at 70 °C for 1 h. Eventually, samples were centrifuged at 2000 rpm for 5 min prior to transferring them into a clean vial insert for GC-MS analysis.

GC-MS analysis was performed on an Agilent 7890 gas chromatograph equipped with a 30-m DB-5MS capillary column with a 10-m Duraguard column connected to an Agilent 5975 MSD operating under electron impact ionisation (Agilent Technologies UK, Ltd.). Samples were injected with an Agilent 7693 autosampler injector into deactivated splitless liners according to the method of Kind et al, 2009, using helium as the carrier gas. Metabolites in the unlabelled pool were identified and quantified using a workflow described by (Behrends et al, 2011). Briefly, samples were deconvoluted in AMDIS (Stein, 1999) and quantified using an in-house script. Integration of labelled metabolites was carried out based on an in-house fragment/retention time database using an updated version of the MatLab script capable of natural isotope correction (Behrends et al, 2011; Tredwell and Keun, 2015).

### Immunocytohistochemistry

Cells were seeded on coverslips in 12-well plates and fixed in 4% formaldehyde for 20 min at room temperature and then washed with 0.2% PBS in Triton (0.2% PBSTr). Following 1 hour blocking with 3% bovine serum albumin in 0.2% PBSTr, cells were incubated with Perilipin-1 (Abcam, ab3526; 1:500) or H4K8ac (1:500; cat. ab15823, Abcam) in blocking buffer for 2 h at room temperature. Following a brief rinse, cells were incubated with a secondary antibody (FITC donkey-anti-rabbit polyclonal, 1:250, Jackson, cat# 711-095-152) for an hour. Subsequently, nuclei were stained with DAPI (1 µg/µL) for 10 min at room temperature, samples were mounted on microscopy slides with Vectashield and visualised using a Zeiss Axiovert 200 M microscope and analysed using densitometry using ImageJ analysis software (NJH).

### Gene expression analysis

Total RNA was extracted and purified from AML12 hepatocytes using an RNeasy Mini Kit (Qiagen) according to the manufacturer's specifications. Purified RNA concentration was quantified at 260 nm using a NanoDrop 100 (Thermo Fisher Scientific).

Each purified RNA sample was diluted with RNase-free water to a final concentration of 100 ng/µL. Complimentary DNA (cDNA) synthesis and genomic DNA elimination in RNA samples was performed using an RT² First Strand Synthesis kit (Qiagen) according to the manufacturer's specifications. The reactions were stored at −20 °C prior to real-time quantitative PCR analysis. The relative abundance of transcripts of interest was measured by qPCR in KAPA SYBR Green (SYBR Green Fast qPCR Master Mix) with a Biorad CFX96 detection system (Applied Biosystems). The SYBR Green qPCR Mastermix contained HotStart DNA Taq Polymerase, PCR Buffer, dNTP mix (dATP, dCTP, dGTP, dTTP) and SYBR Green dye. Before adding cDNA to each well of the 96-well plate, cDNA was diluted in RNase-free water to a final concentration of 8 ng/µL. PCR component mix was prepared by mixing 10 µL SYBR Green qPCR Mastermix with 0.6 µL of 10 µmol/L target primers (forward and reverse; 6 pmoles/reaction) and 4.4 µL RNase-free water. To each well of a 96-well plate, 5 µL cDNA (total amount 40 ng) and 15 µL PCR components mix were added. The plate was centrifuged at 1000×g for 30 s to ensure that the contents were mixed and to remove any bubbles present in the wells. The plate was placed in the real-time cycler with the following cycling conditions: 10 min at 95 °C for 1 cycle to activate HotStart DNA Taq Polymerase; 15 s at 95 °C and 1 min at 60 °C to perform elongation and cooling for 40 cycles. Sequences for qPCR Primers used for mouse *Rn18s*, *Myst-1*, *Gcn5*, *Naa40*, *Naa10*, *Acly*, *Srebf1*, *Fasn* and *Acss2* were purchased from Integrated DNA Technologies (Table 2). Expression levels were normalised to the endogenous control, *Rn18s* for mouse, using the ΔΔCt method and fold changes reported were relative to the control group.

### RNA-sequencing and bioinformatics analysis

Total RNA was isolated from NAA40-KD or negative control cells at 12 and 48 h after siRNA treatment. Sequencing libraries were prepared with poly(A) selection and 150-bp paired-end sequencing on an Illumina NovaSeq by GENEWIZ (South Plainfield, NJ). Using DESeq2, a comparison of gene expression between the customer-defined groups of samples was performed. The false discovery rate test was used to generate *P* values and log2 fold changes. Genes with an adjusted *P* value < 0.05 and absolute log2 fold change >0.5 were called differentially expressed genes.

### Triglyceride assay

Triglyceride measurement was performed using the Triglyceride (TG) Fluorimetric Assay Kit (Cell Biolabs, Inc.) according to the manufacturer's instructions. Weight tissue or cell number values are used to normalise TG measurements.

### Glucose uptake assay

A 2-deoxy-2-[(7-nitro-2,1,3-benzoxadiazol-4-yl) amino]-D-glucose (2-NBDG) glucose uptake assay was performed according to the manufacturer's instructions (Biovision Milpitas, CA). In brief, after starving cells for 6 h with 0.5% FBS-containing DMEM, 2-NBDG was added for 30 min at 37 °C in 5% CO₂. Cells were harvested with trypsin and centrifugation at 1200 rpm for 5 min. The pellet was resuspended in PBS and the fluorescent uptake was measured using

**Table 2. RT-qPCR primer sequence.**

| Gene name | Forward | Reverse |
|---|---|---|
| Mouse *Naa40* | GGGGAGAAAGTCGAGCAAAG | CCCATTGCGGTCATACTTCTTG |
| Human *NAA40* | ATGTAAGCGAGTGTCTGGACT | TGGTTTGCATATTCGTTTTGGTC |
| Mouse *Naa10* | ATGAACATCCGCAATG | ACAATCTTCCCATTCTC |
| Mouse *Gcn5* | GGCTTCTCCGCGAATGACAA | GTTTGGACGCAGCATCTGGA |
| Mouse *Myst-1* | ACGAGGCGATCACCAAAGTG | AAGCGGTAGCTCTTCTCGAAC |
| Mouse *Rn18S* | GCAATTATTCCCCATGAACG | GGCCTCACTAAACCATCCAA |
| Mouse *Acly* | TTCGTCAAACAGCACTTCC | ATTTGGCTTCTTGGAGGTG |
| Mouse *Acss2* | GCTTCTTTCCCATTCTTCGGT | CCCGGACTCATTCAGGATTG |
| Mouse *Srebf* | CCACACTTCATCAAGGCAGA | AGGTACTGTGGCCAAGATGG |
| Mouse *Fasn* | TTGCTGGCACTACAGAATGC | AACAGCCTCAGAGCGACAAT |

flow cytometry. Data were acquired on a BioRad S3e Cell Sorter and analysed using FlowJo software (Treestar).

## Animals

The experimental design and animal management protocols were approved by the local governmental authority responsible for overseeing laboratory animal work and welfare (License number CY/EXP/PR. L1/2019/R1/2022, Ministry of Agriculture and Natural Resources, Republic of Cyprus). All procedures were housed at the animal facility of the University of Cyprus (license number CY.EXP.105). FVB/N wild-type males were used for the purpose of this study. No blinding of animals was performed.

## Induction of hepatic lipid accumulation and treatments

Twelve 8-week-old male mice were randomly divided into two groups: (1) pellet diet (Mucedola, cat # RF25) with tap water (Chow); (2) pellet diet with high fructose/high glucose water (HFHG; 23.1 g L-fructose plus 18.9 g L-glucose in 1 L water). Animals were administered drinking water for each condition for 6 weeks ad libitum and weighed each week. At the end of week 6, mice from each group were further split randomly into four groups: (1) Chow EV ($n = 3$), (2) Chow dO/E ($n = 3$), (3) HFHG EV ($n = 3$), (4) HFHG dO/E ($n = 3$). Animals were injected intraperitonially (200 μL total volume) with 30 μg of each plasmid for co-overexpression of mouse *Gcn5* and human *NAA40* or a cocktail of their corresponding empty vectors (60 μg total/mouse) using PepJet DNA In Vivo Transfection Reagent according to manufacturer's protocol (SignaGen Laboratories). All the animals were sacrificed 48 h post-injection.

## Tissue preservation and sectioning

Mouse livers were excised post-euthanasia. Left and right lobes were snap-frozen individually and stored at −80 °C for further processing. The middle liver lobe was frozen on dry ice to maintain its shape and embedded horizontally in the OCT matrix (CellPath, Ref. KMA-0100-00A). Frozen liver sections (7-μm thick) were obtained with a cryostat (Mainz, SLEE cryostat MEV) and mounted on Superfrost® Plus slides (Thermo Scientific, Ref. J1800AMNZ). Sections were left to airdry (10–15 min, RT) and then were submerged in 1× PBS.

## Lipid staining

Cells were seeded on coverslips in 12-well plates. Cells or tissue sections were washed with 1× PBS and fixed with 4% paraformaldehyde. Intracellular lipids were stained with Nile Red (Invitrogen) as described by the manufacturer. Nuclei/DNA were stained with DAPI (Dako). Images were obtained using a Zeiss Axiovert 200 M microscope and analysed using densitometry using ImageJ analysis software (NJH). Since lipid droplets have different shapes and sizes, Nile red intensity was quantified following background removal. Subsequently, the Nile red intensity was normalised to the number of cells and then each of the conditions was normalised to the average Nile red intensity in the scramble condition.

## Protein lysates

Cell pellets and tissues (50–100 mg) were lysed in 100 μL Cell Extraction Buffer (10 mmol/L Tris, 100 mmol/L NaCl, 1 mmol/L EDTA, 1 mmol/L EGTA, 10 mmol/L NaF, 20 mmol/L $Na_4P_2O_7$, 20 mmol/L $Na_3VO_4$, 1% Triton X-100, 10% glycerol, 0.1% sodium dodecyl sulfate, 0.5% deoxycholate, 1 mmol/L phenylmethylsulfonyl fluoride, complete protease inhibitor tablet and 1% of each phosphatase cocktail inhibitor 2 and 3) for 30 min, vortexing in between at 10-min intervals. The lysate was centrifuged at 13,000× *g* for 10 min at 4 °C and the supernatant was collected and stored at −80 °C.

## Western blotting

The protein concentration was measured by Bradford assay (BioRad). Approximately 20–50 μg of protein was separated on SDS-PAGE and subsequently transferred to a nitrocellulose membrane (GE Healthcare). The membranes were blocked with 5% BSA in TBS-T for 1 h at room temperature and then incubated with respective primary antibodies at 4 °C overnight. The primary antibodies used were: NAA40 (1:1000, ab106408, Abcam), GCN5 (1:1000, sc-365321, Santa-Cruz), ACSS2 (1:1000, ab133664, Abcam), phospho-Akt (Ser473) (1:1000, cat. 9271, Cell signalling), Akt (pan) (1:000, cat. C67E7, Cell Signaling) and β-actin (1:1000; sc-1616-R, Santa-Cruz). The membranes were incubated with the secondary antibody, Horseradish peroxide (HRP)-conjugated goat anti-rabbit IgG (1:30,000, Scientific) or Horseradish peroxide (HRP)-conjugated goat anti-mouse IgG (1:1000, p0447, Dako

Denmark). Bands were visualised by the enhanced chemiluminescence system (BioRad) and analysed using the densitometry tool in ImageJ analysis software (NJH).

## Statistics

Multivariate statistical analyses were performed in GraphPad (GraphPad Prism 9.0; GraphPad Software, San Diego, CA, USA) All variables were log-transformed and subjected to principal component analysis (PCA). The extent to which the model fits and predicts the data is represented by $R^2$ X and $Q^2$ X, respectively.

Data were visualised using GraphPad. All data are expressed as means ± SEM. In GraphPad, unpaired *t* test, one-, or two-way ANOVA was performed where appropriate to determine significant differences between experimental groups. For one-way ANOVA, Dunnett's post hoc multiple comparison tests were performed or a two-stage linear step-up procedure of Benjamini, Krieger and Yekutieli, whilst for two-way ANOVA, Sidak's or Dunnett's post hoc multiple comparison test was used. Outliers were determined by the extreme studentized deviate (ESD) method and excluded. Differences between experimental groups were considered statistically significant when $P \leq 0.05$.

## Data availability

All datasets, including intact lipidomics, MS-based histone acetylations, and transcriptome, generated in this study can be found in Figshare https://doi.org/10.6084/m9.figshare.23257160. The raw mass spectrometry data on histone modifications have been deposited to the ProteomeXchange Consortium (PMID: 24727771) via the PRIDE partner repository with the dataset identifier PXD042624.

## Peer review information

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

## Acknowledgements

This work was funded by a Marie Skłodowska-Curie individual fellowship grant (no. 890750) to E.C, by two EPIC-XS projects (# 0000424 & 0000463) funded by the Horizon 2020 programme of the European Union as well as supported by the European Regional Development Fund and the Republic of Cyprus through the Research & Innovation Foundation (EXCELLENCE/0421/0342, EXCELLENCE/0421/0302 and EXCELLENCE/0421/0152). We would like to thank Professor Julian L. Griffin, Imperial College, for providing LC–MS equipment, and members of the AK laboratory for discussions.

## Author contributions

**Evelina Charidemou**: Conceptualisation; Resources; Data curation; Formal analysis; Funding acquisition; Validation; Investigation; Visualisation; Methodology; Writing—original draft; Writing—review and editing. **Roberta Noberini**: Resources; Formal analysis; Investigation; Methodology; Writing—review and editing. **Chiara Ghirardi**: Methodology; Writing—review and editing. **Polymnia Georgiou**: Methodology; Writing—review and editing. **Panayiota Marcou**: Methodology; Writing—review and editing. **Andria Theophanous**: Methodology; Writing—review and editing. **Katerina Strati**: Resources; Methodology; Writing—review and editing. **Hector Charles Keun**: Resources; Methodology; Writing—review and editing. **Volker Behrends**: Resources; Formal analysis; Investigation; Methodology; Writing—review and editing.

**Tiziana Bonaldi**: Conceptualisation; Resources; Formal analysis; Methodology; Writing—review and editing. **Antonis Kirmizis**: Conceptualisation; Resources; Formal analysis; Supervision; Funding acquisition; Validation; Visualisation; Writing—original draft; Project administration; Writing—review and editing.

## Disclosure and competing interests statement

The authors declare no competing interests.

# Expanded View Figures

**Figure EV1.  Acetyltransferase depletion attenuates insulin signalling and does not affect the transcription of genes involved in lipid synthesis.**

(A) RT-qPCR analysis of expression of *Myst-1*, *Gcn5*, *Naa40*, and *Naa10* mRNA levels in scramble, MYST1-KD, GCN5-KD, NAA40-KD, and NAA10-KD cells, respectively, after 48 h of siRNA treatment. $n = 3$ biological replicates/group. Statistical analysis was performed using two-way ANOVA with post hoc Tukey's multiple-comparisons test; ****$P ≤ 0.0001$. (B) Relative intensity (normalised to scramble) of indicated metabolites in scramble, MYST1-KD, GCN5-KD, NAA40-KD, and NAA10-KD cells measured by MS after 48 h of siRNA treatment. $n = 3$ biological replicates/group. Statistical analysis was performed using a one-way ANOVA with post hoc Dunnett's multiple-comparisons test. Only P values which are *$P ≤ 0.05$ are indicated. All comparisons between KD and Scramble that are non-significant are not indicated.
(C) Relative intensity (normalised to scramble) of free fatty acids and cholesterol in scramble, MYST1-KD, GCN5-KD, NAA40-KD, and NAA10-KD cells measured by MS. $n = 3$ biological replicates/group. Statistical analysis was performed using a one-way ANOVA with post hoc Dunnett's multiple-comparisons test; ns non-significant.
(D) Representative immunoblots of phopshoAKT ser473, totalAKT and β-actin in scramble, MYST1-KD, GCN5-KD, NAA40-KD cells, and NAA10-KD respectively, after 48 h of siRNA treatment. $n = 3$ biological replicates/group. (E) Mean fluorescent uptake quantification by FACS in scramble, MYST1-KD, GCN5-KD, NAA40-KD cells, and NAA10-KD, respectively, after 48 h of siRNA treatment. $n = 3$ biological replicates/group. Statistical analysis was performed using a one-way ANOVA with post hoc Dunnett's multiple-comparisons test; **$P ≤ 0.01$, ns non-significant. (F) RT-qPCR analysis of expression of *Acly*, *Fasn*, *Sbrebf*, mRNA levels in scramble, MYST1-KD, GCN5-KD, NAA40-KD, and NAA10-KD cells, respectively, after 48 h of siRNA treatment. $n = 3$ biological replicates/group. Statistical analysis was performed using a two-way ANOVA with post hoc Tukey's multiple-comparisons test; ns non-significant. (G) Volcano plot comparing mRNA levels between NAA40-KD and SCR control cells as determined by RNA-seq analysis after 12 h of siRNA treatment. Upregulated genes upon loss of NAA40 are shown in red (adjusted $P < 0.05$ and logFC $> 0.5$) and downregulated genes in blue (adjusted $P < 0.05$ and logFC $< -0.5$). $n = 3$ biological replicates/group. Statistical analysis was performed by a paired T test and corrected with the False Discovery Rate (FDR). (H) Volcano plot comparing mRNA levels between NAA40-KD and SCR control cells as determined by RNA-seq analysis after 48 h of siRNA treatment. Upregulated genes upon loss of NAA40 are shown in red (adjusted $P < 0.05$ and logFC $> 0.5$) and downregulated genes in blue (adjusted $P < 0.05$ and logFC $< -0.5$). $n = 3$ biological replicates/group. Statistical analysis was performed by a paired *T* test and corrected with the False Discovery Rate (FDR). (I) Gene ontology analysis of all differentially expressed genes showing enriched biological processes following depletion of NAA40 after 48 h of siRNA treatment. $n = 3$ biological replicates/group. Data information: all data are presented as mean ± SEM.

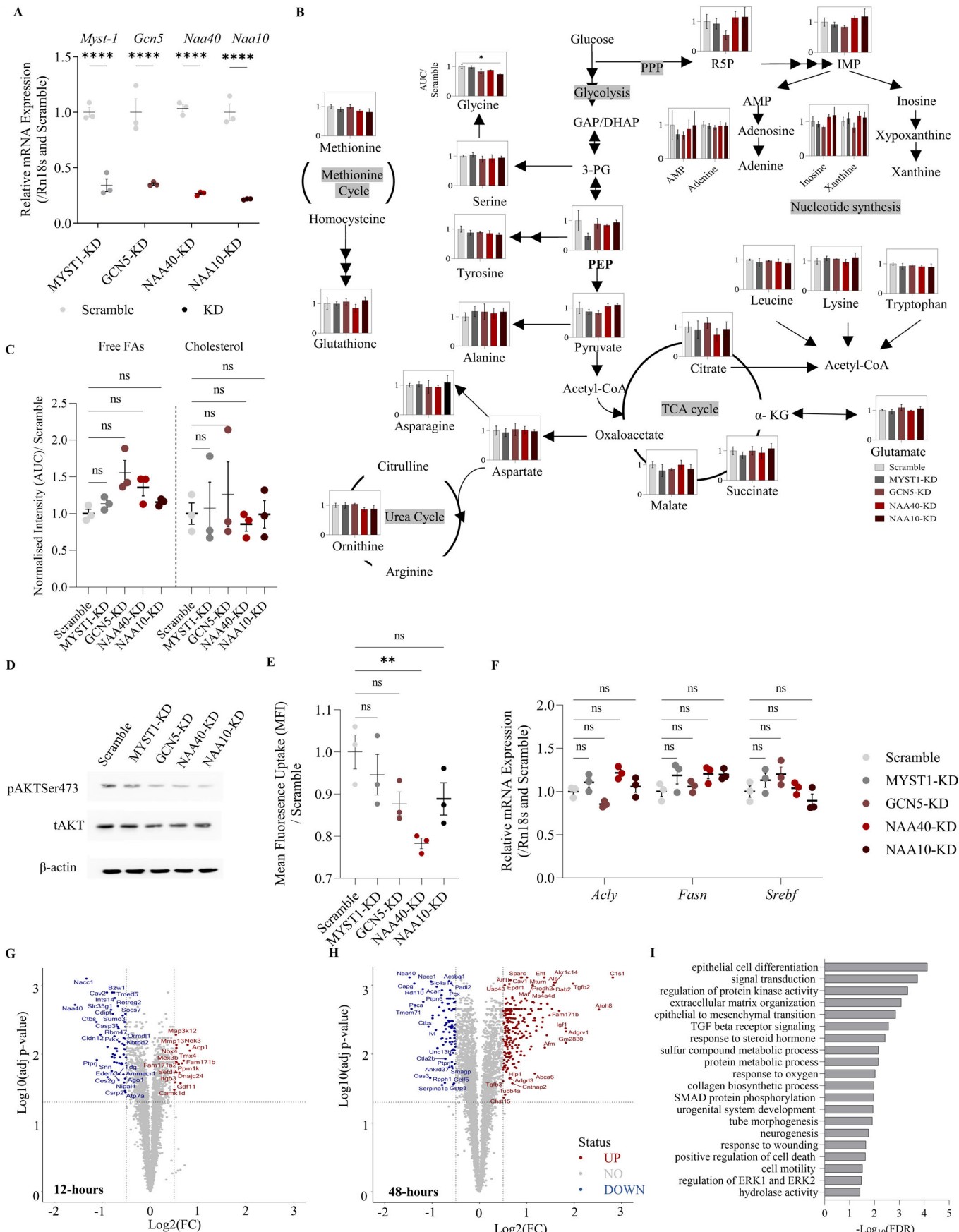

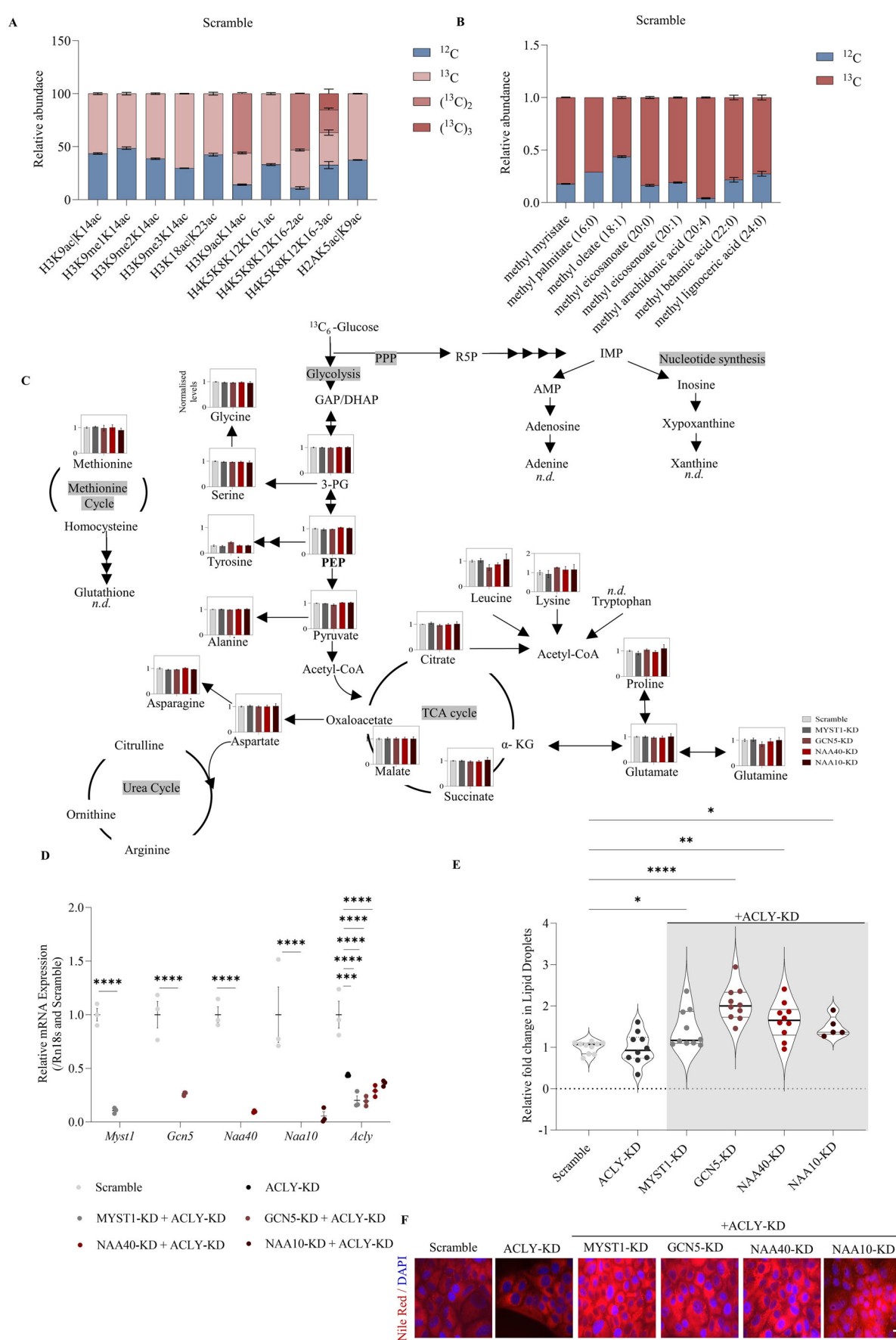

◀

**Figure EV2. Lipid synthesis upon acetyltransferase depletion is not associated with ACLY in AML12 hepatocytes.**

(A) Relative abundance of $^{12}$C and $^{13}$C in histone acetylation peptides measured by MS in scramble cells 48 h after siRNA treatment and supplementation with $^{13}$C$_6$-Glucose. $n = 3$ biological replicates/group. (B) Relative abundance of $^{12}$C and $^{13}$C methyl fatty acids measured by MS in scramble cells 48 h after siRNA treatment and supplementation with $^{13}$C$_6$-Glucose. $n = 3$ biological replicates/group. (C) Normalised levels (% label/ scramble) incorporation in indicated metabolites in scramble, MYST1-KD, GCN5-KD, NAA40-KD, and NAA10-KD cells measured by MS, 48 h after siRNA treatment. $n = 3$ biological replicates/group. Statistical analysis was performed using a one-way ANOVA with post hoc Dunnett's multiple-comparisons test. All comparisons between KD and Scramble that are non-significant are not indicated. (D) RT-qPCR analysis of *Myst-1, Gcn5, Naa40, Naa10* and *Acly* mRNA levels in scramble, ACLY-KD and the indicated double-KD cells after 48 of siRNA treatment. $n = 3$ biological replicates/group. Statistical analysis was performed using two-way ANOVA with post hoc Tukey's multiple-comparisons test; ***$P \leq 0.001$, ****$P \leq 0.0001$. (E) Quantification of relative lipid droplets in scramble, ACLY-KD and the indicated double-KD cells after 48 h of siRNA treatment. $n = 6$–8 biological replicates/group. Statistical analysis was performed using a one-way ANOVA with post hoc Dunnett's multiple-comparisons test; *$P \leq 0.05$, **$P \leq 0.01$, ****$P \leq 0.0001$. (F) Representative images of lipid droplets Nile red (red) and nuclei by DAPI (blue) of scramble, ACLY-KD and the indicated double-KD cells after 48 h of siRNA treatment. Scale bar = 25 μm. $n = 6$–8 biological replicates/group. Data information: (A–D) are presented as mean ± SEM; (E) is presented as a violin plot.

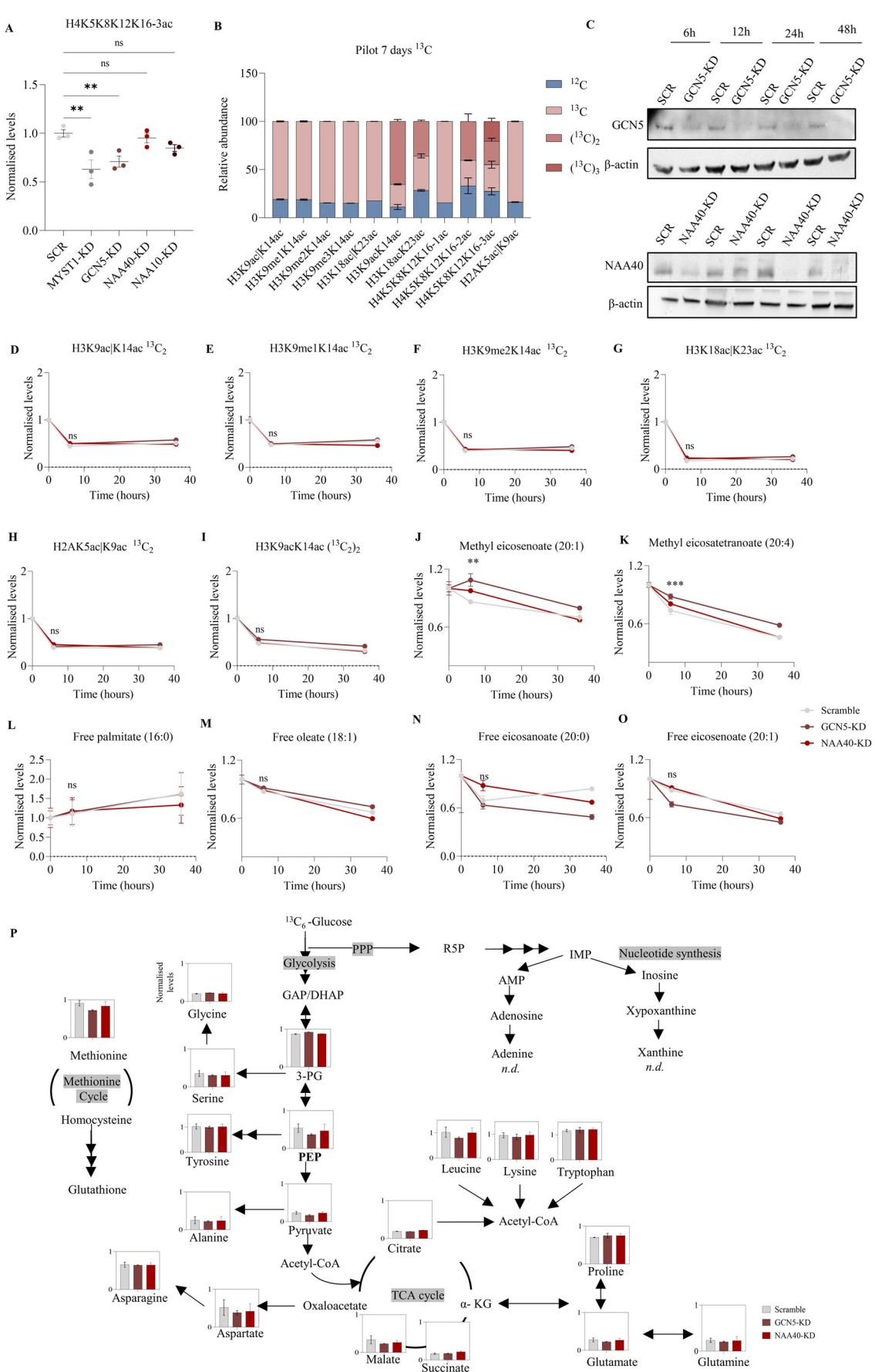

◀ **Figure EV3. Analysis of histone acetylation marks, methyl fatty acids, free fatty acids and aqueous metabolites upon acetyltransferase depletion in chromatin-enriched $^{13}$C-labelled cells.**

(A) Normalised levels (%RA/scramble) of the triply acetylated H4K5K8K12K16 peptide in scramble, MYST1-KD, GCN5-KD, NAA40-KD, and NAA10-KD cells measured by MS, 48 h after siRNA treatment. $n = 3$ biological replicates/group. Statistical analysis was performed using a one-way ANOVA with post hoc Dunnett's multiple-comparisons test; **$P \leq 0.01$, ns non-significant. (B) Relative abundance of $^{12}$C and $^{13}$C in acetylated histone peptides in AML12 cells 7 days after supplementation with $^{13}$C$_6$-Glucose. $n = 3$ biological replicates/group. (C) Representative immunoblots of GCN5, NAA40 and β-actin in scramble, GCN5-KD, and NAA40-KD cells at 6, 12, 24 and 48 h after siRNA treatment. $n = 3$ biological replicates/group. (D) Normalised levels (%label 6 h/%label 0 h) of singly labelled H3K9ac|K14ac in scramble, GCN5-KD, and NAA40-KD cells. $n = 3$ biological replicates/group. Statistical analysis was performed using two-way ANOVA with post hoc Tukey's multiple-comparisons test; ns non-significant. (E) Normalised levels (%label 6 h/%label 0 h) of singly labelled H3K9me1K14ac in scramble, GCN5-KD, and NAA40-KD cells. $n = 3$ biological replicates/group. Statistical analysis was performed using two-way ANOVA with post hoc Tukey's multiple-comparisons test; ns non-significant. (F) Normalised levels (%label 6 h/%label 0 h) of singly labelled H3K9me2K14ac in scramble, GCN5-KD, and NAA40-KD cells. $n = 3$ biological replicates/group. Statistical analysis was performed using two-way ANOVA with post hoc Tukey's multiple-comparisons test; ns non-significant. (G) Normalised levels (%label 6 h/%label 0 h) of singly labelled H3K18ac | K23 in scramble, GCN5-KD, and NAA40-KD cells. $n = 3$ biological replicates/group. Statistical analysis was performed using two-way ANOVA with post hoc Tukey's multiple-comparisons test; ns non-significant. (H) Normalised levels (%label 6 h/%label 0 h) of singly labelled H2AK5ac|K9ac in scramble, GCN5-KD, and NAA40-KD cells. $n = 3$ biological replicates/group. Statistical analysis was performed using two-way ANOVA with post hoc Tukey's multiple-comparisons test; ns non-significant. (I) Normalised levels (%label 6 h/%label 0 h) of doubly labelled H3K9acK14ac in scramble, GCN5-KD, and NAA40-KD cells in scramble, GCN5-KD, and NAA40-KD cells. $n = 3$ biological replicates/group. Statistical analysis was performed using two-way ANOVA with post hoc Tukey's multiple-comparisons test; ns non-significant. (J) Normalised levels (%label 6 h/%label 0 h) of methyl eicosenoate (20:1) in scramble, GCN5-KD, and NAA40-KD cells. $n = 3$ biological replicates/group. Statistical analysis was performed using two-way ANOVA with post hoc Tukey's multiple-comparisons test; **$P \leq 0.01$. (K) Normalised levels (%label 6 h/%label 0 h) of Methyl eicosatetranoate (20:4) in scramble, GCN5-KD, and NAA40-KD cells. $n = 3$ biological replicates/group. Statistical analysis was performed using two-way ANOVA with post hoc Tukey's multiple-comparisons test; ***$P \leq 0.001$. (L) Normalised levels (%label 6 h/%label 0 h) of free palmitate (16:0) in scramble, GCN5-KD, and NAA40-KD cells. $n = 3$ biological replicates/group. Statistical analysis was performed using two-way ANOVA with post hoc Tukey's multiple-comparisons test; ns non-significant. (M) Normalised levels (%label 6 h/%label 0 h) of free oleate (18:1) in scramble, GCN5-KD, and NAA40-KD cells. $n = 3$ biological replicates/group. Statistical analysis was performed using two-way ANOVA with post hoc Tukey's multiple-comparisons test; ns non-significant. (N) Normalised levels (%label 6 h/%label 0 h) of free eicosanoate (20:0) in scramble, GCN5-KD, and NAA40-KD cells. $n = 3$ biological replicates/group. Statistical analysis was performed using two-way ANOVA with post hoc Tukey's multiple-comparisons test; ns non-significant. (O) Normalised levels (%label 6 h/%label 0 h) of free eicosenoate (20:1) in scramble, GCN5-KD, and NAA40-KD cells. $n = 3$ biological replicates/group. Statistical analysis was performed using two-way ANOVA with post hoc Tukey's multiple-comparisons test; ns non-significant. (P) Normalised levels (%label 6 h/%label 0 h) of the indicated aqueous metabolites in scramble, GCN5-KD, and NAA40-KD cells. $n = 3$ biological replicates/group. Statistical analysis was performed using a one-way ANOVA with post hoc Dunnett's multiple-comparisons test. All comparisons between KD and Scramble that are non-significant are not indicated. Data information: all data are presented as mean ± SEM. Source data are available online for this figure.

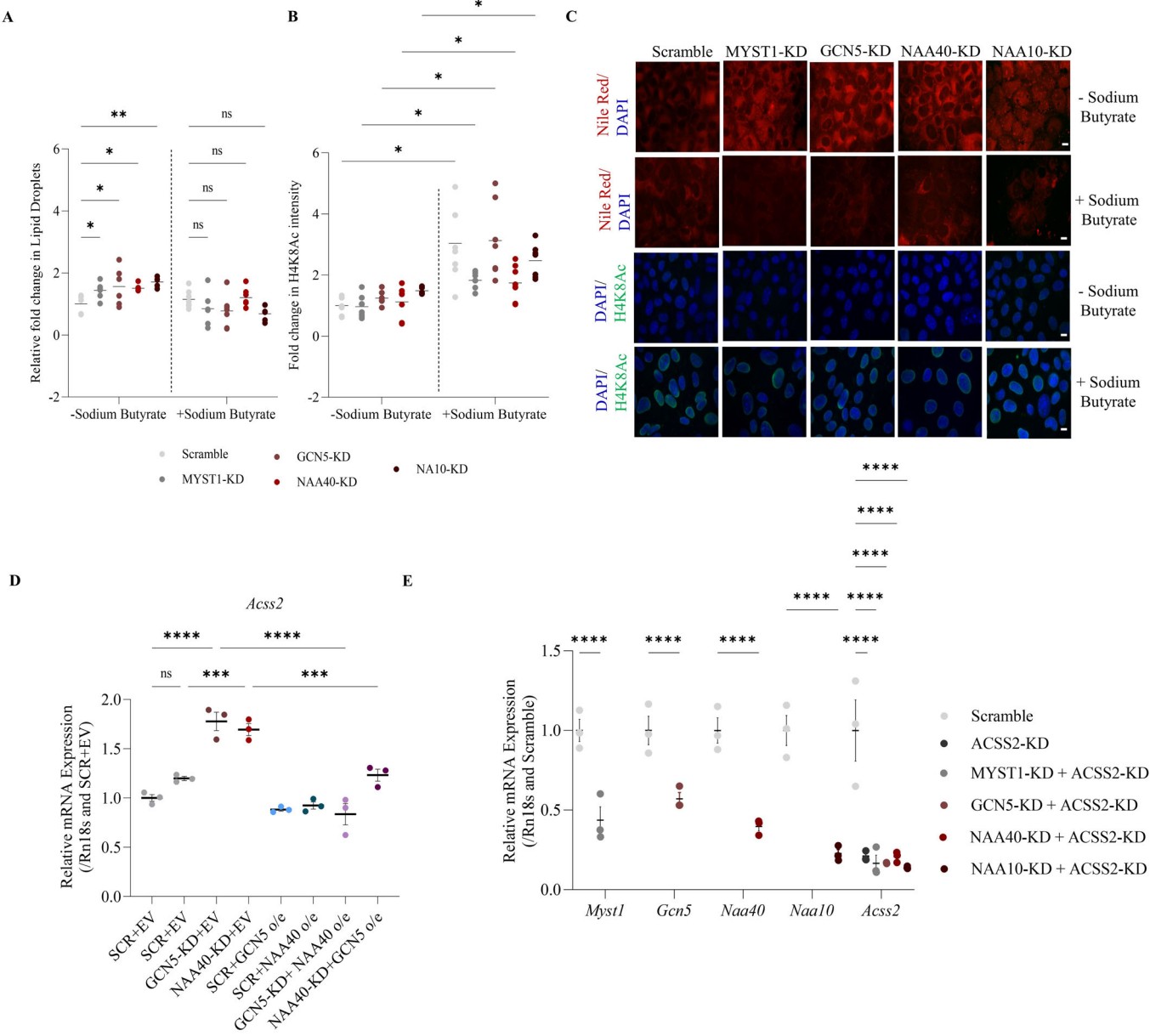

**Figure EV4. Lipid synthesis upon acetyltransferase depletion is associated with ACSS2-driven HDAC-dependent acetate formation.**

(A) Quantification of H4K8Ac using ImageJ MYST1-KD, GCN5-KD, NAA40-KD, and NAA10-KD cells with and without sodium butyrate, 48 h after siRNA treatment. $n = 6$ biological replicates/group. Statistical analysis was performed using two-way ANOVA with post hoc Tukey's multiple-comparisons test; *$P \leq 0.05$, **$P \leq 0.01$, ns non-significant. (B) Quantification of relative lipid droplets in scramble, MYST1-KD, GCN5-KD, NAA40-KD, and NAA10-KD cells with and without sodium butyrate, 48 h after siRNA treatment. $n = 6$ biological replicates/group. Statistical analysis was performed using two-way ANOVA with post hoc Tukey's multiple-comparisons test. Statistical analysis was performed using two-way ANOVA with post hoc Tukey's multiple-comparisons test; *$P \leq 0.05$. (C) Representative images of lipid droplets by Nile red staining (red), H4K8Ac (green) and nuclei by DAPI (blue) in scramble, MYST1-KD, GCN5-KD, NAA40-KD, and NAA10-KD cells with and without sodium butyrate, 48 h after siRNA treatment; Scale bar = 25 μm. $n = 6$ biological replicates/group. (D) RT-qPCR analysis of *Acss2* mRNA levels in the indicated treatment groups. $n = 3$ biological replicates/group. Statistical analysis was performed using a one-way ANOVA with post hoc Dunnett's multiple-comparisons test; ***$P \leq 0.001$, ****$P \leq 0.0001$, ns non-significant. (E) RT-qPCR analysis of *Myst-1, Gcn5, Naa40, Naa10* and *Acss2* mRNA levels in scramble, ACSS2-KD and the indicated double-KD cells after 48 h of siRNA treatment. $n = 3$ biological replicates/group. Statistical analysis was performed using two-way ANOVA with post hoc Tukey's multiple-comparisons test. ****$P \leq 0.0001$. Data information: all data are presented as mean ± SEM.

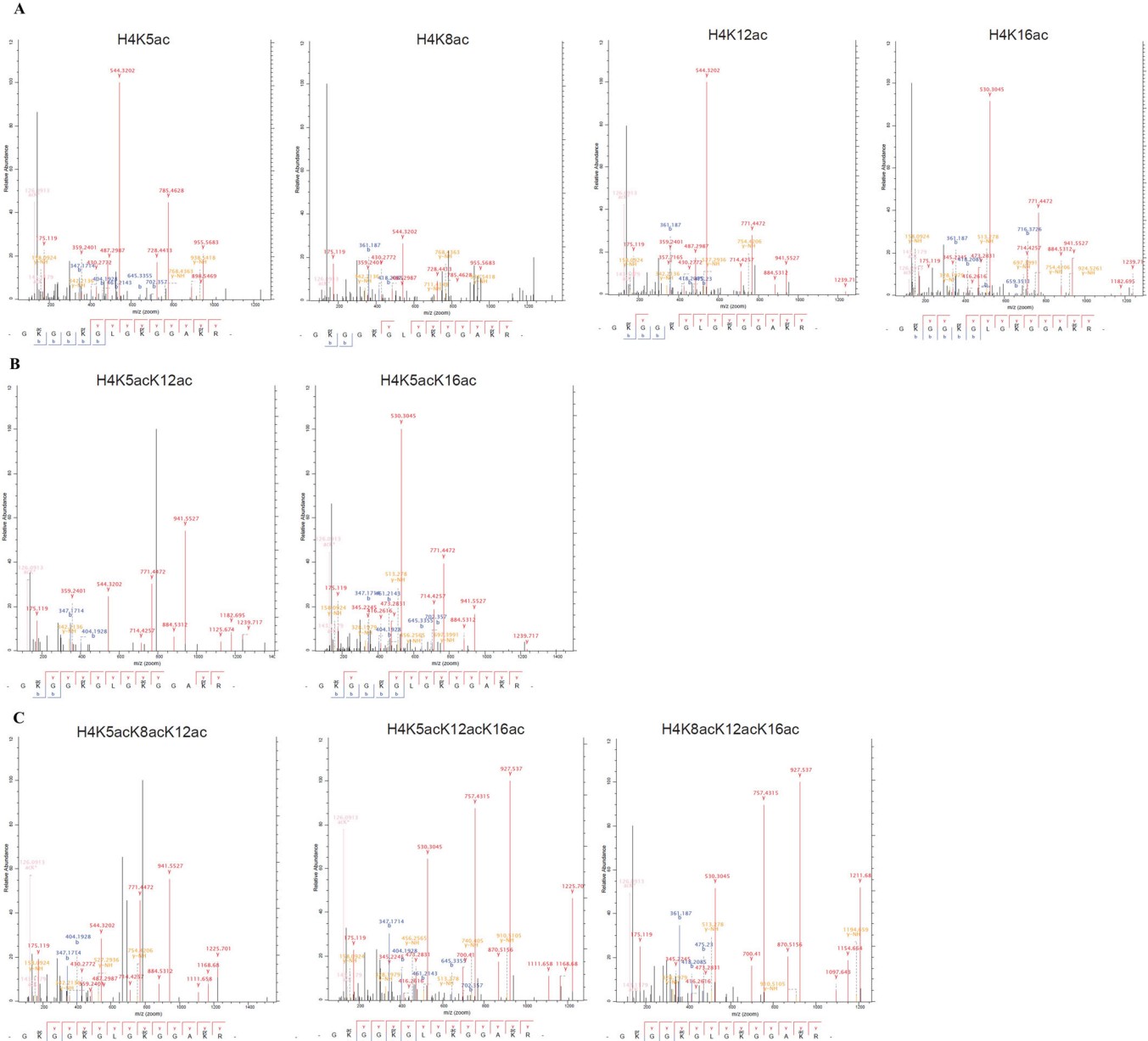

**Figure EV5.  MS-MS spectra for the differentially acetylated forms of the histone H4 tail.**

(A) MS-MS spectra for differentially acetylated H4K5K8K12K16-1ac histone peptide. (B) MS-MS spectra for differentially acetylated H4K5K8K12K16-2ac histone peptide. (C) MS-MS spectra for differentially acetylated H4K5K8K12K16-3ac histone peptide.

