## [Peer Review File · The EMBO Journal]

Hyperacetylated histone H4 is a source of carbon contributing to lipid synthesis

Evelina Charidemou, Roberta Noberini, Chiara Ghiradi, Polymnia Georgiou, Panayiota Marcou, Andria Theofanous, Katerina Strati, Hector Keun, Volker Behrends, Tiziana Bonaldi, and Antonis Kirmizis

Corresponding author(s): Antonis Kirmizis (kirmizis@ucy.ac.cy)

Review Timeline:

Submission Date:	26th Jun 23
Editorial Decision:	15th Sep 23
Revision Received:	6th Dec 23
Editorial Decision:	10th Jan 24
Revision Received:	12th Jan 24
Accepted:	31st Jan 24

Editor: Cornelius Schneider

Transaction Report:

Dear Dr. Kirmizis,

Thank you for submitting your manuscript for consideration by the EMBO Journal and for sharing a preliminary revision plan with me.

As can be seen from the reports, two out of three referees found the results of high importance and interest, and all three referees agreed that the experiments were performed competently but require major revisions to substantiate the claims raised in the manuscript. We have also considered the concerns regarding the novelty of the findings by referee #3. We feel that there is overall a strong interest in the community to study the interconnections between histone modifications and cellular metabolism, and have therefore decided that these novelty concerns do not have to be considered when preparing the revised manuscript. Given that this manuscript revisits previous findings, it is of particular importance that the technical concerns raised by the referees are addressed convincingly.

Based on these considerations and your willingness to engage in a major revision as indicated during the pre-decision consultation, I would like to invite you to submit a revised version of the manuscript, addressing the comments of all three reviewers. I should add that it is EMBO Journal policy to allow only a single round of revision, and acceptance of your manuscript will therefore depend on the completeness of your responses in this revised version. If you have any additional questions or want to discuss the revisions further, I am happy to do so by email or video conferencing.

We generally allow three months as standard revision time, which can be extended to 6 months in case of major revisions, such as the experiments required here. As a matter of policy, competing manuscripts published during this period will not negatively impact on our assessment of the conceptual advance presented by your study. However, we request that you contact the editor as soon as possible upon publication of any related work, to discuss how to proceed. Should you foresee a problem in meeting the deadline, please let us know in advance and we may be able to grant an extension.

Thank you for the opportunity to consider your work for publication. I look forward to your revision.

Yours sincerely,

Cornelius Schneider

Cornelius Schneider, PhD
Editor
The EMBO Journal
c.schneider@embojournal.org

- a point-by-point response to the referees' comments, with a detailed description of the changes made (as a word file).
- a word file of the manuscript text.

- individual production quality figure files (one file per figure)
- a complete author checklist, which you can download from our author guidelines (<https://www.embopress.org/page/journal/14602075/authorguide>).
- Expanded View files (replacing Supplementary Information)
Please see out instructions to authors
<https://www.embopress.org/page/journal/14602075/authorguide#expandedview>

We realize that it is difficult to revise to a specific deadline. In the interest of protecting the conceptual advance provided by the work, we recommend a revision within 3 months (14th Dec 2023). Please discuss the revision progress ahead of this time with the editor if you require more time to complete the revisions. Use the link below to submit your revision:

Referee #1:

The manuscript by Charidemou et al. describes how histone acetyltransferases sequester acetyl-CoA/acetate on histones, thereby limiting their availability for ACSS2-dependent lipogenesis. The main conceptual advance of this paper is showing that non-specific H4 histone acetylation is a reservoir for carbon availability for lipid metabolism, in contrast to prior studies looking at histone deacetylation. Overall, this is a comprehensive and multidisciplinary paper and the conclusions are of interest. The work would benefit from a few control experiments and technical considerations.

1. To quantify lipid accumulation, the authors stain cells with Nile Red, perform microscopy, and measure lipid droplets/cell number. Can the authors further explain this quantification method? How can there be only 1 lipid droplet/cell? While the representative images look like there may be a difference, more precise quantification would greatly strengthen the story. Additionally, a lipid extraction and triglyceride measurement in all the conditions tested, or at least the more important ones in addition to Fig. 1H, would complement the microscopy work and further verify that there is increased lipid accumulation in HAT KD cells since the microscopy work often has a low n=6 cells.
2. In Figure 1H, it looks like there is a small difference in the triglycerides. Is this difference statistically significant? This finding would also complement data in Figure 5. Does giving the cells a HFHG media induce a larger difference in total TG? In figure 5, increasing the sample size for 5b/c would further corroborate these data.
3. When is HAT expression induced? Is HAT expression regulated by nutrient availability or fasting state? Does Gcn5, Naa40 or Myst mRNA or protein expression increase when wildtype mice are given HFHG water compared to regular water?
4. An isotope-labeled lipid synthesis assay with a lipid extraction and triglyceride measurement with ACSS2 KD, HAT KD cells would show that functionally, this mechanism is ACSS2-dependent.
5. Why do the control and HFHG mice gain weight at similar rates? I would expect the HFHG mice to gain weight more quickly than the saline control. Furthermore, it would help the in vivo data to have a lipid extraction and triglyceride measurement from mouse liver. Or, the authors can try IP injection prior to HFHG water start to test whether these mice are resistant to diet-induced lipid accumulation.
6. In Fig. EV1, the authors say there is no difference in lipid synthesis gene expression, but in Fig. 4, there are differences in Acss2 mRNA expression. Can the authors comment? Are there differences in the Srebp1c lipogenic pathway by qPCR? The question of which came first, the chicken or egg also comes up: Does histone acetylation directly affect the gene expression of Acss2 or is it non-specifically the acetyl-CoA availability that then drives lipogenesis? Is Acly expression different?

Minor Comments

1. Acetyl-CoA synthetase activity vs. protein expression are two separate things, so that might be more accurate to say in the text. Similarly, lipid synthesis is different from lipid accumulation.

2. In Figure 5f, it might be more appropriate to say cells with HFHG media, instead of HFHG diet since they are cells in vitro. Also, in Figure 7, it might be more appropriate to say HFHG water rather than diet since each group eats the same chow diet.

Referee #2:

The work from Charidemou and coworkers interrogates the role of acetylated histones as acetyl-CoA reservoir. The work stems from the observation (both published data from the Kirmizis group and new data shown here) that HATs knock down promotes lipid synthesis in a way that is possibly independent from HAT-specific transcriptional programs. The data indicate that acetyl-CoA moieties are released from the chromatin and used for lipid synthesis, consistent with recent reports (Hsieh, 2022; Mendoza, 2022) although others claim a negligible contribution (Soaita, 2023). The topic is clearly debated and there is vibrant interest around it, so there is clear need for additional data. The authors do a good job combining mass spectrometry approaches (for lipid and histone modifications quants) as well as both in vitro and in vivo systems that reflect high anabolic states. This provides a framework to possibly reconcile contradicting evidence, where chromatin-bound carbon atoms are used for lipid synthesis to fulfill elevated anapldrotic demands.

The work is high level and suitable for publication in EMBO J, although I have a number of concerns that shall be addressed.

1) generally speaking, after Fig 1, authors use lipid droplets as a readout for lipid synthesis. While the two are often linked, this is not always the case. Key experiments that demonstrate enhanced channeling of acetyl-CoA for lipid synthesis should use lipid synthesis as readout indeed (isotopologue analysis, % labeling, etc).

2) to the very same point, authors do a good job in figure 1 demonstrating that lipid deposition in HAT KD cells seems independent from nutrient uptake, but evidence is not conclusive. This is an extremely important point that should be nailed down. First, is lipid catabolism altered? Is FAO regulated by HAT KD? Is autophagy impaired? Second, can lipids be generated from alternative carbon stores (i.e.: glycogen)?? I believe authors should get to these points before really drawing any conclusion.

3) I liked the experiments in Fig 3&4, which are pivotal. However, ¹³C atoms that are used for de novo lipid synthesis can come from non-chromatin carbon stores that have accumulated over 7 days in culture, including glycogen, amino acids, possibly nucleotides. Authors should at least demonstrate that labeling increase is still there upon ACLY KD and possibly abolished by ACSS2 KD. The exp in Fig 4F&G are OK but far from conclusive. Moreover, they use LD as readout (see concern #1) and the difference claimed for 4G is not clear to me.

4) I was puzzled by the experiments in Fig 5, where authors claim to reproduce findings of Zhao, 2020 using in vitro settings. In fact, authors culture hepatocytes in high fructose medium and observe enhanced lipogenesis. However, the Nature paper reported negligible fructolysis and fructose was shown to minimally contribute to the acetyl-CoA pool. I wonder is this might be an effect of ACSS2 up regulation. Is the experiment performed in acetate-free conditions (dialyzed serum)? More in general, experiment in Fig 5&6 are really intriguing and might provide a novel framework to the field, as I said at the beginning. For this reason, I would invite the authors to confirm some of the data they obtain from histone MS, which are interesting but can sometimes give false positives.

Minor issues:

- biplots in Fig 1 are really hard to read and interpret. Can the authors graph the data in different way?
- is ACSS2 differentially acetylated in your systems?
- 6B is not clear
- panel numbers are mixed up in the text from time to time

Referee #3:

Charidemou et al describe a series of experiments investigate how acetylated histones might be a source of acetyl-groups for lipid synthesis. The authors deplete certain histone acetyltransferases (HATs) and observed enhanced lipid synthesis. The effect appears to be dependent on deacetylases and the acetyl-CoA synthase ACSS2. Overexpression of some HATs in hepatocytes can reverse diet-induced lipogenesis. The authors conclude that "this study highlights histone acetylation as metabolite reservoir that can directly contribute carbon to lipid synthesis." While there are some interesting observations, in some ways the data seem preliminary for the stated claims, and yet the overall idea that acetate from histones is recycled to acetyl-CoA is not novel. Below is a non-exhaustive list of concerns:

1.) It has been known since the 1960s (ACETYLATION AND METHYLATION OF HISTONES AND THEIR POSSIBLE ROLE IN THE REGULATION OF RNA SYNTHESIS* BY V. G. ALLFREY, R. FAULKNER, AND A. E. MIRSKY, PNAS 1964) that free radiolabeled acetate is incorporated into histones, both in the backbone and as acetylgroups on lysines. From the 1990-2000s, we also know that ACSS2 is likely the major enzyme responsible for converting this acetate to acetyl-CoA. Acetyl-CoA generated from acetate is then available for lipid synthesis or other acetyl-CoA reactions. This reviewer is finding it difficult to see what is new.

2.) It seems like the most novel part might be the competition between HATs and lipid synthesis, but here it is not clear if the HATs are altering the chromatin state and that is what leads to a change in lipid gene expression. This aspect is underdeveloped. Other consumers of acetyl-CoA could be utilized to test this hypothesis.

3.) The tracing experiments don't give actual levels of acetyl-groups and stoichiometry. It is difficult to assess if the level of an

acute deacetylation can maintain lipid synthesis without new acetylation-deacetylation cycles.

4.) The section "Hyperacetylated H4 is a carbon source for lipid synthesis upon HAT-depletion" is a convoluted set of experiments. As designed, these experiments are very challenging to interpret for a number of reasons. It appears that amino-acid ^{13}C labeling was not accounted for, and it was assumed that all labels are from acetate. It is unclear why under these conditions a large amount of deacetylated H4 is the main source of carbon for lipid synthesis. This makes little sense and the data presented are not compelling. Of course, if free acetate is available and ACSS is expressed, then some of these carbons will end up on some lipids, but how much relative to non-histone sources?

Referee #1 (Report for Author)

The manuscript by Charidemou et al. describes how histone acetyltransferases sequester acetyl-CoA/acetate on histones, thereby limiting their availability for ACS2-dependent lipogenesis. The main conceptual advance of this paper is showing that non-specific H4 histone acetylation is a reservoir for carbon availability for lipid metabolism, in contrast to prior studies looking at histone deacetylation. Overall, this is a comprehensive and multidisciplinary paper and the conclusions are of interest. The work would benefit from a few control experiments and technical considerations.

We would like to express our gratitude to the reviewer for their positive assessment of our work and their recognition of the central conceptual breakthrough. Specifically, we appreciate their acknowledgement of the findings that histone acetyltransferases play a role in sequestering acetyl-CoA/acetate on histones, thereby influencing ACS2-dependent lipogenesis.

1. To quantify lipid accumulation, the authors stain cells with Nile Red, perform microscopy, and measure lipid droplets/cell number. Can the authors further explain this quantification method? How can there be only 1 lipid droplet/cell? While the representative images look like there may be a difference, more precise quantification would greatly strengthen the story. Additionally, a lipid extraction and triglyceride measurement in all the conditions tested, or at least the more important ones in addition to Fig. 1H, would complement the microscopy work and further verify that there is increased lipid accumulation in HAT KD cells since the microscopy work often has a low $n=6$ cells.

We thank the reviewer for bringing up this quantification issue and apologize for the confusion. Firstly, we would like to clarify that the n number in the figure legends corresponds to biological replicates, i.e., images are from at least 6 different biological replicates, and we show a representative image for each condition. Since lipid droplets have different shapes and sizes, we quantify by measuring Nile red intensity following background removal. Subsequently, the Nile red intensity is normalized to the number of cells and then each of the conditions is normalized to the average Nile red intensity in the scramble condition. Therefore, the y-axis represents the relative fold change of lipid droplet (which is normalized to the average value found in scramble cells). We have now included the description of the quantification procedure in the methods section (Lines 650-653) and corrected the y-axis of all image plots showing "Fold Change in lipid droplets".

We agree with the reviewer's feedback and therefore to further strengthen our findings, we have included triglyceride quantifications for the following key experiments:

- Figure 4D. Triglyceride quantification for double KD (ACS2+HAT-KD conditions)
- Figure 5D. Triglyceride quantification for control and HFHG-supplementation in empty vector and double overexpression conditions in vitro
- Figure 6F. Triglyceride quantification for control and HFHG-supplementation in empty vector and double overexpression conditions in vivo

2. In Figure 1H, it looks like there is a small difference in the triglycerides. Is this difference statistically significant? This finding would also complement the data in Figure 5. Does giving the cells a HFHG media induce a larger difference in total TG? In Figure 5, increasing the sample size for 5b/c would further corroborate these data.

We would like to thank the reviewer for their observation. The change in Figure 1H is indeed subtle, but it holds statistical significance. We have now included a triglyceride quantification for the high

fructose and glucose experiments (Fig 5D & 6F), where the change in triglyceride content is even more pronounced in mouse livers further corroborating the data in Figure 1H.

3. When is HAT expression induced? Is HAT expression regulated by nutrient availability or fasting state? Does Gcn5, Naa40 or Myst mRNA or protein expression increase when wildtype mice are given HFHG water compared to regular water?

Thank you for raising this point. To address this question, we have included Western blot analyses assessing the protein levels of acetyltransferases in HFHG conditions for both the in vitro and in vivo study. Even though there is an increase in the in vitro study, this is not corroborated by the in vivo study as the protein levels of both GCN5 and NAA40 are not changed significantly with HFHG water supplementation (Figs 5E and 6G).

4. An isotope-labelled lipid synthesis assay with a lipid extraction and triglyceride measurement with ACSS2 KD, HAT KD cells would show that functionally, this mechanism is ACSS2-dependent.

We greatly appreciate the reviewer's insight. We have performed an additional labelling experiment, where we enriched cells with ¹³C and subsequently conducted a double knockdown of either GCN5 or NAA40 acetyltransferase together with ACSS2. We then collected data for the time point at which we previously observed an increase in the label incorporation into lipids. With this additional labelling experiment, we show that the label is retained on the histone fraction and is not enriched in lipids, indicating that ACSS2 is needed for lipid induction upon acetyltransferase depletion (Fig 4F-I, lines 250-260).

In addition, as mentioned above in point 1, we have also quantified triglycerides in these double-KD conditions for all acetyltransferases, showing again that an increase in triglycerides upon acetyltransferase depletion is lost when ACSS2 is also depleted simultaneously (Fig. 4D).

5. Why do the control and HFHG mice gain weight at similar rates? I would expect the HFHG mice to gain weight more quickly than the saline control. Furthermore, it would help the in vivo data to have a lipid extraction and triglyceride measurement from mouse liver. Or, the authors can try IP injection prior to HFHG water start to test whether these mice are resistant to diet-induced lipid accumulation.

We appreciate this insightful comment. It is worth noting that the effects of the high fructose and glucose-supplemented water on weight tend to become more pronounced around the 6th week, as supported by Softic et al 2017 *JCI* (PMID: 28972537), which is similar to our observations. Taking this into account, we strategically selected week 6 for IP injections. This timing ensures that we are operating within a window where the observed phenotype is still potentially reversible, enhancing the overall robustness of our experimental approach.

As stated in point 1, we conducted a triglyceride assay on liver tissues (Fig. 6F), corroborating the increase in liver triglyceride content with HFHG supplementation, that is lost with acetyltransferase overexpression.

The reviewer's comment on performing IP injections prior to the feeding study is insightful. However, the overexpression through this IP injection method is transient and therefore, it cannot last for the entire period of the feeding study.

6. In Fig. EV1, the authors say there is no difference in lipid synthesis gene expression, but in Fig. 4, there are differences in *Acss2* mRNA expression. Can the authors comment? Are there differences in the *Srebp1c* lipogenic pathway by qPCR? The question of which came first, the chicken or egg also comes up: Does histone acetylation directly affect the gene expression of *Acss2* or is it non-specifically the acetyl-CoA availability that then drives lipogenesis? Is *Acly* expression different?

We thank the reviewer for bringing up these issues. In relation to gene expression, *Acss2* mRNA levels do not change after 12 hours of NAA40 KD (Fig EV1G). However, after 48 hours of NAA40-KD, *Acss2* mRNA levels increased almost significantly both in the RNA-seq analysis (Fig EV1H) and qPCR analysis (Fig 4A). *Acss2* mRNA and protein levels are also significantly changing in the absence of all other acetyltransferases (Fig 4A-B). This increased *Acss2* expression possibly indicates a response to an overall acetylation/deacetylation imbalance upon acetyltransferase depletion, with histones being more deacetylated. We included text in Discussion (lines 356-360), discussing this change in *Acss2* expression. In terms of whether histone acetylation directly affects *Acss2* expression, we would expect that in the absence of the specific HATs, we would not get an increase in their specific histone acetylation marks at this locus. However, we cannot exclude the possibility that other acetylation marks or other histone modifications are changing at the *ACSS2* locus, but examining this possibility would be a major advancement in itself and therefore beyond the scope of this work.

Despite the changes observed on *Acss2* levels, in general, we do not see any significant changes in lipid synthesis genes, including *srebp1*. We have performed qPCR analysis to monitor the expression of various lipid synthesis genes including *Acly*, *Fasn* and *Srebf* and did not observe any major differences between scramble cells and acetyltransferase-depleted cells (Fig. EV1F), consistent with the RNA-seq analysis (Fig EV1 G-H).

The question of 'what comes first' is valid. Our data support the hypothesis that acetyl-CoA availability drives lipogenesis. Firstly, in our previous paper (Charidemou et al, PMID: 35057804) we observed an increase in lipids already at 6 hours post knockdown, which is before *Acss2* expression changes (i.e. 12 hours post acetyltransferase-KD). Secondly, the depletion of different acetyltransferases, targeting histone or non-histone proteins, has the same effect on lipogenesis supporting that the effect of acetyltransferase depletion occurs regardless of their specific roles on chromatin states. This possibility is discussed in the Discussion section (lines 330-360).

Minor Comments

1. Acetyl-CoA synthetase activity vs. protein expression are two separate things, so that might be more accurate to say in the text. Similarly, lipid synthesis is different from lipid accumulation.

Based on the reviewer's comment we have modified the text accordingly (i.e. lines 29, 94, 145, 200, 264, 281, 299-300, 306, 319-321, 340, 405)

2. In Figure 5f, it might be more appropriate to say cells with HFHG media, instead of HFHG diet since they are cells in vitro. Also, in Figure 7, it might be more appropriate to say HFHG water rather than diet since each group eats the same chow diet.

We agree and we have modified both figures 5 and 6 as well as their corresponding text (lines 264-306).

Referee #2 (Report for Author)

The work from Charidemou and coworkers interrogates the role of acetylated histones as acetyl-CoA reservoir. The work stems from the observation (both published data from the Kirmizis group and new data shown here) that HATs knockdown promotes lipid synthesis in a way that is possibly independent from HAT-specific transcriptional programs. The data indicate that acetyl-CoA moieties are released from the chromatin and used for lipid synthesis, consistent with recent reports (Hsieh, 2022; Mendoza, 2022) although others claim a negligible contribution (Soaita, 2023). The topic is clearly debated and there is vibrant interest around it, so there is clear need for additional data. The authors do a good job combining mass spectrometry approaches (for lipid and histone modifications quants) as well as both in vitro and in vivo systems that reflect high anabolic states. This provides a framework to possibly reconcile contradicting evidence, where chromatin-bound carbon atoms are used for lipid synthesis to fulfill elevated anapldrotic demands.

The work is high level and suitable for publication in EMBO J, although I have a number of concerns that shall be addressed.

We thank the reviewer for the constructive feedback and their recognition of the valuable contributions this work makes to the field, rendering it well-suited for publication in EMBO.

1) generally speaking, after Fig 1, authors use lipid droplets as a readout for lipid synthesis. While the two are often linked, this is not always the case. Key experiments that demonstrate enhanced channeling of acetyl-CoA for lipid synthesis should use lipid synthesis as readout indeed (isotopologue analysis, % labeling, etc).

This point complements the feedback from reviewer #1. To further enhance our findings concerning lipid droplet accumulation, we performed a triglyceride quantification assay for double KD experiments involving each acetyltransferase together with ACSS2. The results revealed lack of triglyceride accumulation (Fig. 4D), supporting the requirement of ACSS2 in this process. Furthermore, % labelling was employed as an additional lipid analysis within double KD cells (Fig. 4I). Moreover, we measured triglyceride levels in HFHG experiments, demonstrating that the increase in triglyceride levels due to HFHG supplementation was reduced by acetyltransferase overexpression, both in vitro (Fig. 5D) and in vivo (Fig. 6F), consistent with the lipid droplet readout.

2) to the very same point, authors do a good job in figure 1 demonstrating that lipid deposition in HAT KD cells seems independent from nutrient uptake, but evidence is not conclusive. This is an extremely important point that should be nailed down. First, is lipid catabolism altered? Is FAO regulated by HAT KD? Is autophagy impaired? Second, can lipids be generated from alternative carbon stores (i.e.: glycogen)?? I believe authors should get to these points before really drawing any conclusion.

We thank the reviewer for raising this point. Our dataset includes free fatty acid data and methyl ester fatty acids derived from distinct lipids. More specifically, our data supports that there are no changes in cholesterol, free fatty acid content (Fig. EV1C) or flux of ¹³C to free fatty acids (Fig. EV3L-O) supporting that lipid catabolism is not altered. We have also included a statement about these findings in the main text (lines 139-142 & 223-227, respectively).

Furthermore, our analysis extended to aqueous metabolites demonstrating that other catabolic pathways (such as Krebs cycle, amino acids metabolism and glycolysis which are pathways changing in autophagy) also remained unaffected (Fig EV1B, lines 130-132).

3) I liked the experiments in Fig 3&4, which are pivotal. However, ¹³C atoms that are used for de novo lipid synthesis can come from non-chromatin carbon stores that have accumulated over 7 days in culture, including glycogen, amino acids, possibly nucleotides. Authors should at least demonstrate that labeling increase is still there upon ACLY KD and possibly abolished by ACSS2 KD. The exp in Fig 4F&G are OK but far from conclusive. Moreover, they use LD as readout (see concern #1) and the difference claimed for 4G is not clear to me.

We thank the reviewer for this important point. We have now included analysis of our labelling datasets which shows that in free fatty acids, amino acids, and glycolytic intermediates we did not observe changes in ¹³C enrichment upon acetyltransferase depletion (Figs EV2C & EV3P and described in lines 189-191 & 225-226).

Furthermore, we have now performed a labelling experiment upon double knockdown of either GCN5 or NAA40 acetyltransferase together with ACSS2 and we find that the label is retained onto histones and not enriched in lipids (Fig. 4F-I, and lines 250-260), validating that ACSS2 is needed for lipid induction upon acetyltransferase depletion.

Lastly, we performed a triglyceride assay to quantitatively assess lipid changes for double KD hepatocytes and demonstrated that lipid enrichment is abolished for all double KD cells (Fig. 4D, lines 248-250), consistent with the lipid droplet readout.

4) I was puzzled by the experiments in Fig 5, where authors claim to reproduce findings of Zhao, 2020 using in vitro settings. In fact, authors culture hepatocytes in high fructose medium and observe enhanced lipogenesis. However, the Nature paper reported negligible fructolysis and fructose was shown to minimally contribute to the acetyl-CoA pool. I wonder if this might be an effect of ACSS2 up regulation. Is the experiment performed in acetate-free conditions (dialyzed serum)? More in general, experiment in Fig 5&6 are really intriguing and might provide a novel framework to the field, as I said at the beginning. For this reason, I would invite the authors to confirm some of the data they obtain from histone MS, which are interesting but can sometimes give false positives.

We thank the reviewer for raising these issues. The findings of Zhao et al were mainly used as a guide where lipogenesis could be induced in an ACSS2-dependent manner that is irrelevant to ACLY. In particular, we implemented a high fructose **and** glucose diet that can contribute to more acetyl-CoA when combined. We have rephrased the text accordingly to make it clear that we used this as a tool for lipogenesis induction (lines 264-270). Importantly, the experiments were performed in dialyzed FBS in order to ensure that carbon atoms are not derived from free acetate in FBS.

In relation to the MS data, we believe that this is the most accurate and quantitative method for analyzing histone modifications. The differences identified in this study, although reproducible and statistically significant, are mild, and hardly detectable with classical antibody-based methods. In addition, available antibodies recognizing histone acetylations cannot discriminate among peptides containing one or multiple modifications and would be useless in our context, where we are specifically interested in hyper-acetylated peptides. An anti H4K5ac antibody has been in some instances used as a proxy of hyper-acetylation (because H4K5 is supposed to be the last of the four

histone H4 lysine residue to be acetylated, according with the hypothesis of a “zip” model whereby acetylation proceeds in the direction of from K16 to K5 (PMID: 12239278). However, in our MS data we detect H4K5 acetylation also in mono- and di-acetylated histone H4 peptides, therefore we cannot use this antibody. We have looked into designing antibodies that would recognize hyperacetylated histones (i.e. concomitantly detecting 3 or 4 acetylated lysine sides), but unfortunately this is not a viable option. We have now added example MS/MS spectra for acetylated peptides examined in this study (Fig. EV5).

Minor issues:

- biplots in Fig 1 are really hard to read and interpret. Can the authors graph the data in different way? We have now plotted only PCA, where there is clear discrimination between scramble cells and acetyltransferase depleted cells for both lipids and lipid-bound fatty acids. Higher positive values of PC1 indicate more lipids (Fig. 1A) and more lipid-bound fatty acids (Fig. 1B).
- is ACSS2 differentially acetylated in your systems? Although this is an interesting question, we did not monitor the acetylation of ACSS2 as we cannot find a readily available anti-acetylated ACSS2 antibody. In any case, this investigation would be a major advancement in itself and therefore beyond the scope of this paper.
- 6B is not clear → We modified the figure so that we now demonstrate the change in weight after the IP injection.
- panel numbers are mixed up in the text from time to time → thank you for noticing, we have made corrections in the text.

Referee #3 (Report for Author)

Charidemou et al describe a series of experiments investigate how acetylated histones might be a source of acetyl-groups for lipid synthesis. The authors deplete certain histone acetyltransferases (HATs) and observed enhanced lipid synthesis. The effect appears to be dependent on deacetylases and the acetyl-CoA synthase ACSS2. Overexpression of some HATs in hepatocytes can reverse diet-induced lipogenesis. The authors conclude that "this study highlights histone acetylation as metabolite reservoir that can directly contribute carbon to lipid synthesis." While there are some interesting observations, in some ways the data seem preliminary for the stated claims, and yet the overall idea that acetate from histones is recycled to acetyl-CoA is not novel. Below is a non-exhaustive list of concerns:

1.) It has been known since the 1960s (ACETYLATION AND METHYLATION OF HISTONES AND THEIR POSSIBLE ROLE IN THE REGULATION OF RNA SYNTHESIS* BY V. G. ALLFREY, R. FAULKNER, AND A. E. MIRSKY, PNAS 1964) that free radiolabeled acetate is incorporated into histones, both in the backbone and as acetylgroups on lysines. From the 1990-2000s, we also know that ACSS2 is likely the major enzyme responsible for converting this acetate to acetyl-CoA. Acetyl-CoA generated from acetate is then available for lipid synthesis or other acetyl-CoA reactions. This reviewer is finding it difficult to see what is new.

We appreciate the reviewer's comments and assessment of this study. We agree that the idea of acetate being incorporated into histones and the fact that ACSS2 is responsible for converting acetate into acetyl-CoA have been previously demonstrated (as we cite many of these papers in our text). However, this is the first study that demonstrates that histone-derived acetyl-CoA contributes to lipid synthesis. As noted by the other reviewers above, this is a debated topic and these additional data that we provide will benefit the topic that chromatin-bound carbon can fuel metabolic pathways (see comments from referee 2). An additional conceptual advance of this work is that nonspecific acetylation, in addition to the previously demonstrated deacetylation, affects the carbon availability from the chromatin reservoir for lipid metabolism (see comments from referee 1). Finally, this study provides novel evidence that a specific form of hyperacetylated histone H4 is the source of these carbon atoms.

2.) It seems like the most novel part might be the competition between HATs and lipid synthesis, but here it is not clear if the HATs are altering the chromatin state and that is what leads to a change in lipid gene expression. This aspect is underdeveloped. Other consumers of acetyl-CoA could be utilized to test this hypothesis.

We thank the reviewer for this insightful comment. The evidence so far in the study supports lipid enrichment upon depletion of acetyltransferases independently of transcriptional changes (EV1G-H). However, based on the reviewer's comment, to further strengthen this point we have now introduced new data on the protein acetyltransferase NAA10, which is not known to have any histone substrates [Aksnes et al, 2019 Mol Cell, PMID: 30878283], as an alternative consumer of acetyl-CoA. Specifically, we demonstrate that NAA10 depletion:

- Increased lipid and lipid-bound fatty acid content as well as lipid droplet formation, without altering lipid breakdown or any other key metabolic pathways, and did not change lipid synthesis gene expression, similarly to the other depleted histone acetyltransferases (Fig. 1 & EV1).

- Likewise to other depleted acetyltransferases, increased the levels of 13C-hyperacetylated H4 peptide when cells were supplemented with 13C-glucose upon NAA10-depletion (such as fig. 2B-C).
- decreased levels of bulk 12C-hyperacetylated H4, again corroborating the other 3 depleted histone acetyltransferases (Fig.3A & EV3A).
- along with ACSS2 depletion diminished triglyceride content and lipid droplet accumulation, showing ACSS2 dependency like the other acetyltransferases (Fig. 4A-E) and independently of ACLY (EV2D-F).
- diminished lipid droplet accumulation upon HDAC inhibition with sodium butyrate (Fig.EV4A-C)

This additional research significantly reinforces the novelty of this study, illustrating that beyond hyperacetylated histone H4 serving as a metabolite reservoir, this effect is not limited to histone acetyltransferases but extends to acetyl-CoA-consuming acetyltransferases in general. This evidence proposes that the relative activity of acetyltransferases and deacetylases can have an impact on chromatin reservoir thereby affecting lipid synthesis. We have incorporated this information into the discussion section (lines 338-343).

3.) The tracing experiments don't give actual levels of acetyl-groups and stoichiometry. It is difficult to assess if the level of an acute deacetylation can maintain lipid synthesis without new acetylation-deacetylation cycles.

We agree with the reviewer that this is an important point. However, this is a major advancement that will require an in-depth investigation and therefore beyond the scope of this study.

4.) The section "Hyperacetylated H4 is a carbon source for lipid synthesis upon HAT-depletion" is a convoluted set of experiments. As designed, these experiments are very challenging to interpret for a number of reasons. It appears that amino-acid 13C labeling was not accounted for, and it was assumed that all labels are from acetate. It is unclear why under these conditions a large amount of deacetylated H4 is the main source of carbon for lipid synthesis. This makes little sense and the data presented are not compelling. Of course, if free acetate is available and ACSS is expressed, then some of these carbons will end up on some lipids, but how much relative to non-histone sources?

This point resonates with the insights provided by reviewers #1 and #2. As mentioned above, we have now included analysis of our datasets on various aqueous metabolites, encompassing glycolytic intermediates, TCA cycle metabolites, and amino acids. These data indicate that there are no changes in the flux of 13C to other metabolic pathways beyond its initial context (Fig.EV2C & EV3P).

In relation to the last point, we would like to emphasize that these experiments were performed using dialyzed FBS (acetate-free condition) to eliminate this possibility.

Dear Dr. Kirmizis,

Thank you for submitting your manuscript for consideration by the EMBO Journal. It has now been seen by referees #1 and #2 whose comments are enclosed. As you will see, both the referees' express interest in your manuscript and are broadly in favor of publication.

However, referee #2 still voices concerns regarding your response to the major concern 2. While we agree with these concerns, we also realize that experiments and data reanalysis here is in line with your proposed revisions in your preliminary point-by-point response which this referee deemed sufficient previously. We would therefore not require additional experiments to address this point but would ask you to cover these limitations in the discussion.

To the same point this referee also raises another concern regarding the lack of % lipid analysis in the HATs KD and ACLY/ACSS2 KD experiments. Could you please comment on this point. Again, I find the revision experiments to be in line with the preliminary point-by-point response but would appreciate your arguments here.

In addition, I would have a number of editorial points which I would like to ask you to address before submitting a revised version of the manuscript.

- Our data editors have indicated that the following needs to be corrected/added in the figure legends:

1. Please note that the figure 4c-e does not contain any statistical test parameter, kindly rectify the statistics related information in the figure legend appropriately.
2. Please note that the figure 4c does not contain a micrograph, kindly rectify the scale bar related information in the figure legend appropriately.
3. The statistical test related information for legend 5f is incorrectly labelled as 5e. This needs to be rectified.
4. The error bar definition in the legend for figure 5f is incorrectly labelled as 5g. This needs to be rectified."
5. Please indicate the statistical test used for data analysis in the legends of figures 4a, g-i; 5a, d, g; 6c, g-h; EV 2e.
6. Please note that in figures 2c-d; 5c-d, f; 6e-g; EV 1a, c, e-f; EV 4a-b, d-e, there is a mismatch between the annotated p values in the figure legend and the annotated p values in the figure file that should be corrected.
7. Please note that for the figures EV 2a-c; EV 3a-b, d-p; p-values and statistical tests are indicated in the legends. However, comparison for the same, ""*****/***/**/*"" has not been represented in the figures. Please rectify this in the figures or legends as applicable."
8. Please note that information related to n is missing in the legends of figure 6h; EV 3a.
9. Although 'n' is provided, please describe the nature of entity for 'n' in the legends of figures 1c, e-h; 2c-d; 3c-n; 4a-d, g-i; 5c-d, f-g; EV 1a-c, e-h; EV 2a-e; EV 3a-b, d-p; EV 4a-b, d-e."
10. There seems to be a possible re-use of image in Figure EV2 F / MYST1-KD & NAA40-KD.
11. The Western blot in Figure EV3 C - NAA40 is very tightly cropped and possibly cut off. Could you please provide source data for this western blot.

We generally allow three months as standard revision time. As a matter of policy, competing manuscripts published during this period will not negatively impact on our assessment of the conceptual advance presented by your study. However, we request that you contact the editor as soon as possible upon publication of any related work, to discuss how to proceed.

Thank you for the opportunity to consider your work for publication. I look forward to your revision.

Yours sincerely,

Cornelius Schneider

Cornelius Schneider, PhD
Editor
The EMBO Journal
c.schneider@embojournal.org

We realize that it is difficult to revise to a specific deadline. In the interest of protecting the conceptual advance provided by the work, we recommend a revision within 3 months (9th Apr 2024). Please discuss the revision progress ahead of this time with the editor if you require more time to complete the revisions. Use the link below to submit your revision:

Referee #1:

The authors have been responsive to the concerns raised. I now support publication.

Referee #2:

The authors presented a revised version of their manuscript the documents the release of carbon moieties from acetylated histones to facilitate lipid biosynthesis.

The manuscript shows significant improvements, mostly to Figure 4. Authors addressed all my concerns, which is very commendable, although some points were not necessarily hammered down.

I don't think my point that other macromolecular reservoirs might be used for lipid synthesis was tackled. Authors perform metabolomic analysis (including aqueous polar metabolites) that shows no alterations upon HATs KD and conclude catabolism was not impacted. But compensatory fluxes might take place to restore metabolic homeostasis. Ultimately, I am not sure authors chose the best experiment to prove their point, which remains somehow incompletely addressed. Results certainly align

with authors' hypothesis, but I thought this issue was critical and I am not sure the revised manuscript makes a much stronger point.

To the same point, authors nicely demonstrated that ACSS2 KD abrogates many effects linked to HATs KD. However, somehow they did not show the most direct experiments (% lipid labeling - as in Fig 4F - upon HATs KD and ACLY/ACSS2 KD. This is puzzling because all other evidence, which is substantial and technically sound, feels much more circumstantial.

I was asked to comment on authors response to reviewer #3. While I don't agree with some of reviewer's initial comments (about novelty, "convoluted" experimental design, i.e.) I believe authors do a good job in their response. I liked the NaButyrate experiment. I already commented on the robustness of polar metabolomic and its significance for the study.

Referee #1:

The authors have been responsive to the concerns raised. I now support publication. We would like to thank the reviewer for the positive response.

Referee #2:

The authors presented a revised version of their manuscript the documents the release of carbon moieties from acetylated histones to facilitate lipid biosynthesis.

The manuscript shows significant improvements, mostly to Figure 4. Authors addressed all my concerns, which is very commendable, although some points were not necessarily hammered down.

I don't think my point that other macromolecular reservoirs might be used for lipid synthesis was tackled. Authors perform metabolomic analysis (including aqueous polar metabolites) that shows no alterations upon HATs KD and conclude catabolism was not impacted. But compensatory fluxes might take place to restore metabolic homeostasis. Ultimately, I am not sure authors chose the best experiment to prove their point, which remains somehow incompletely addressed. Results certainly align with authors' hypothesis, but I thought this issue was critical and I am not sure the revised manuscript makes a much stronger point.

This constitutes a valid point that has been incorporated into our discussion. Despite our integration of data pertaining to the relative levels and ¹³C fluxes of various aqueous metabolites, we acknowledge the potential existence of additional compensatory fluxes aimed at reinstating homeostasis. This discussion point can be found on lines 390-393.

To the same point, authors nicely demonstrated that ACSS2 KD abrogates many effects linked to HATs KD. However, somehow they did not show the most direct experiments (% lipid labeling - as in Fig 4F - upon HATs KD and ACLY/ACSS2 KD. This is puzzling because all other evidence, which is substantial and technically sound, feels much more circumstantial.

Thank you for pointing this out. Figure 4I demonstrates Normalized levels (%label 6h/ %label 0h) of indicated methyl fatty acids (which corresponds to the fatty acids shown in figure 3 and EV3) in scramble, ACSS2-KD, and double KD cells. We have now changed the names of the x-axis to make it clear that these are lipid species.

I was asked to comment on authors response to reviewer #3. While I don't agree with some of reviewer's initial comments (about novelty, "convoluted" experimental design, i.e.) I believe authors do a good job in their response. I liked the NaButyrate experiment. I already commented on the robustness of polar metabolomic and its significance for the study.

We would like to thank the reviewer for the positive feedback.

Dear Dr. Kirmizis,

I am pleased to inform you that your manuscript has been accepted for publication in the EMBO Journal.

Yours sincerely,

Cornelius Schneider, PhD
Editor
The EMBO Journal
c.schneider@embojournal.org
